# Oxidative phosphorylation is a metabolic vulnerability of endocrine therapy and palbociclib resistant metastatic breast cancers

Rania El-Botty [1], Ludivine Morriset[1], Elodie Montaudon[1], Zakia Tariq[2], Anne Schnitzler[2], Marina Bacci [3], Nicla Lorito[3], Laura Sourd[1], Léa Huguet[1], Ahmed Dahmani[1], Pierre Painsec[1], Heloise Derrien[1], Sophie Vacher [2], Julien Masliah-Planchon[2], Virginie Raynal[4], Sylvain Baulande [4], Thibaut Larcher [5], Anne Vincent-Salomon [6], Guillaume Dutertre[7], Paul Cottu [8], Géraldine Gentric[9], Fatima Mechta-Grigoriou [9], Scott Hutton [10], Keltouma Driouch [2], Ivan Bièche[2,11], Andrea Morandi [3] & Elisabetta Marangoni [1] ✉

Resistance to endocrine treatments and CDK4/6 inhibitors is considered a near-inevitability in most patients with estrogen receptor positive breast cancers (ER + BC). By genomic and metabolomics analyses of patients' tumours, metastasis-derived patient-derived xenografts (PDX) and isogenic cell lines we demonstrate that a fraction of metastatic ER + BC is highly reliant on oxidative phosphorylation (OXPHOS). Treatment by the OXPHOS inhibitor IACS-010759 strongly inhibits tumour growth in multiple endocrine and palbociclib resistant PDX. Mutations in the *PIK3CA/AKT1* genes are significantly associated with response to IACS-010759. At the metabolic level, in vivo response to IACS-010759 is associated with decreased levels of metabolites of the glutathione, glycogen and pentose phosphate pathways in treated tumours. In vitro, endocrine and palbociclib resistant cells show increased OXPHOS dependency and increased ROS levels upon IACS-010759 treatment. Finally, in ER + BC patients, high expression of OXPHOS associated genes predict poor prognosis. In conclusion, these results identify OXPHOS as a promising target for treatment resistant ER + BC patients.

Oestrogen receptor positive breast cancers (ER + BC) account for over 80% of primary breast malignancies[1]. Classically ER + BC patients are treated with endocrine therapies (ET), which block ER-signalling and significantly increase patients' survival. However, resistance to ET remains a significant problem, and a large proportion of patients display cancer recurrence and distant metastases during or after adjuvant treatment[2]. Inhibitors of cyclin-dependent kinases 4 and 6 (CDK4/6) have been recently shown to improve progression-free and overall survival[3] and are now the standard of care for the treatment of advanced ER + BC. However, intrinsic or acquired resistance to CDK4/6 inhibitors is a frequent event, limiting the success of these treatments[4,5]. Identifying novel therapies for the treatment of CDK4/6 inhibitor-resistant patients is of great importance.

Treatment resistance and development of metastases are associated to transcriptional and metabolic reprogramming[6–10]. The metabolic reprogramming supports the different steps of the metastatic process, including intravasation, the ability to survive when in circulation and invasion at the distant organs[9]. At the preclinical level, studies on cell lines showed that luminal and basal-like breast cancer cell lines show different metabolic profiles associated with different metastatic characteristics[11].

Studies with patient-derived xenografts (PDX) of triple-negative BC showed that lung metastases displayed enrichment of a gene expression signature of mitochondrial oxidative phosphorylation (OXPHOS) as compared to the matched primary tumours (PT)[12]. Although these results are based on transcriptional changes, they suggest that metastases are metabolically different as compared with their corresponding PT. Evidence of a metabolic rewiring based on metabolite concentration analyses is less demonstrated due to technical issues and the scarce amount of tissue in metastases biopsies.

In this work, we develop different PDX models of ER + BC metastases that provide the opportunity to perform both transcriptomic and metabolic studies of matched primary tumours and metastases-derived samples. Through these analyses, we identify an up-regulation of OXPHOS and mitochondrial metabolic pathways in the metastases-derived samples and validate it as a therapeutic target by treating different PDX models with an OXPHOS inhibitor. By genomic analysis, we identify amplification of *MYC* and mutation of *PIK3CA* or *AKT1* genes in the responding tumours. Moreover, by performing a pharmacodynamics metabolomics analysis of PDX responding differentially to the OXPHOS inhibitor, we identify metabolic changes associated with the response. Finally, in the last part of the manuscript, we analyse the expression of different OXPHOS genes and their prognostic significance in tumours from a cohort of 503 breast cancer patients with 20 years of follow-up.

## Results

### Transcriptomic and metabolomics changes in bone metastases
From a set of bone metastasis (BM) derived PDX, established from ER + BC patients and recently described (Fig. 1A), we identified by comparative gene set enrichment analysis (GSEA) different pathways enriched in BM samples as compared to matched primary tumours (PT)[13]. OXPHOS, a metabolic pathway that occurs in mitochondria to produce adenosine triphosphate (ATP), was among the top enriched hallmarks in BM PDX (Fig. 1B)[13].

To further analyse the metabolic reprogramming in BM-derived samples, we performed a global metabolomics analysis of six BM PDX (five were described in the previous study and one PDX was newly established) and six patients' breast PT (four out of the six pairs were from matched patients/PDX), by ultra-performance liquid chromatography (UPLC) coupled with mass spectroscopy (MS). The hierarchical clustering of breast PT and BM PDX samples is shown in Fig. 1C. BM PDX samples show up-regulation of many metabolites associated to the major metabolic pathways. Among the top enriched pathways, we found gamma-glutamyl amino acids, the TCA cycle, sugar metabolism and purine/pyrimidine metabolisms (Fig. 1D). The heat map of the statistically significant biochemical entities differentially represented between PT and BM PDX are shown in Supplementary Data 1.

The levels of several gamma-glutamyl amino acids, that play an important role in glutathione homeostasis through the gamma-glutamyl cycle, are shown in Fig. 1E and are significantly enriched in BM PDX. Moreover, the levels of the relevant TCA intermediates are also enriched, as shown in Fig. 1F in BM PDX. Although the levels of the increased metabolites revealed by the steady state analysis do not imply per se enhancement of the TCA flux, an overall increase in the TCA cycle intermediates could contribute to the production of reducing equivalents (NADH + H+ and FADH2) and subsequent activity of the mitochondrial electrons transport respiratory chain[14].

Glucose can be diverted into the hexosamine pathway, which generates nucleotide sugars for glycosylation, an enzymatic process that produces glycosidic linkages of saccharides to other saccharides, proteins, or lipids. Here, hexosamine pathway metabolites such as glucosamine 6-phosphate, UDP-glucose, UDP-galactose, and UDP-glucuronate were elevated in BM PDX samples compared to breast PT (Supplementary Data 1 and Fig. 1G), suggesting increased glycosylation might occur in these samples. Different metabolites of purine and pyrimidine metabolism were increased in BM PDX models (Fig. 1H), reflecting the increased proliferation and DNA replication identified in the GSEA analysis and previously described[13].

### Targeting OXPHOS inhibits tumour growth in multiple PDX models of ER-positive BM
Based on the findings from transcriptomic and metabolomics analyses, we hypothesised that OXPHOS could represent a therapeutic target in metastatic ER + BC. Therefore, we tested the antitumour activity of IACS-010759 (hereafter called IACS), an inhibitor of the complex I of the electron respiratory chain[15] in 7 PDX of ER + BM (five of these models were included in the previous metabolomics analysis). Of the seven PDX exposed to IACS, five responded with tumour regression or stable disease, whereas two were resistant (Fig. 2A and Fig. S1A). The PDX showing the highest response to IACS (HBCx-124) also responded to the 5 mg/kg dose, while the 2.5 mg/kg dose did not impair tumour growth significantly (Fig. 2B). By contrast to IACS, metformin, an antidiabetic drug that also inhibits mitochondrial complex I[16], did not decrease tumour growth in this model (Fig. 2B). Of note, among the responding PDX, 3 were resistant to fulvestrant (Fig. 2A).

To decipher the biological mechanisms and pathways affected by IACS treatment, we performed an RNAseq analysis of HBCx-124 xenografts after 1 week of IACS treatment. A gene set enrichment analysis (GSEA) of gene expression level in treated and untreated HBCx-124 permit to identify downregulation of different gene sets associated with cell proliferation, E2F targets, G2M checkpoint and mitosis in IACS-treated xenografts. Up-regulated gene sets were associated with activation of the mRNA upon binding of the cap-binding complex, translation initiation and cell death (Fig. 2C and Table S1).

Next, we asked whether targeting OXPHOS would be effective in PDX established from other different types of metastases, either from another subtype of BC or from another organ. We evaluated the efficacy of IACS in 2 PDX established from TNBC BM, one PDX established from an ER-negative liver metastasis (whose patient's PT was ER+) and two PDX established from ER+ primary BC. All these additional PDX were resistant to IACS treatment (Fig. S1B).

Collectively, these results demonstrate that targeting OXPHOS is a metabolic vulnerability in metastatic ER + BC, including endocrine-resistant tumours.

### Oxidative phosphorylation is a metabolic vulnerability in PDX models with primary and secondary resistance to the CDK4/6 inhibitor palbociclib
CDK4/6 inhibitors have been approved as first-line treatment for advanced BC patients in combination with aromatase inhibitors or fulvestrant. However, most patients develop resistance to CDK4/6 inhibitors and finding new therapeutic targets for these patients remains an unmet clinical need. We, therefore, developed and characterised two PDX models of acquired resistance to palbociclib (HBCx-124 PalboR25 and HBCx-134 PalboR31) described recently[13]. The GSEA analysis of HBCx-124 PalboR25 (Fig. 2D) revealed enrichment of different pathways associated with mitochondrial metabolism in the palboR as compared to the parental palbo-S, including OXPHOS (M5936), the respiratory electron transport (M893) and Mitochondrial transcription factor A (TFAM) target genes (M30304) (Fig. 2E). Next, we evaluated the antitumour efficacy of IACS either alone or in

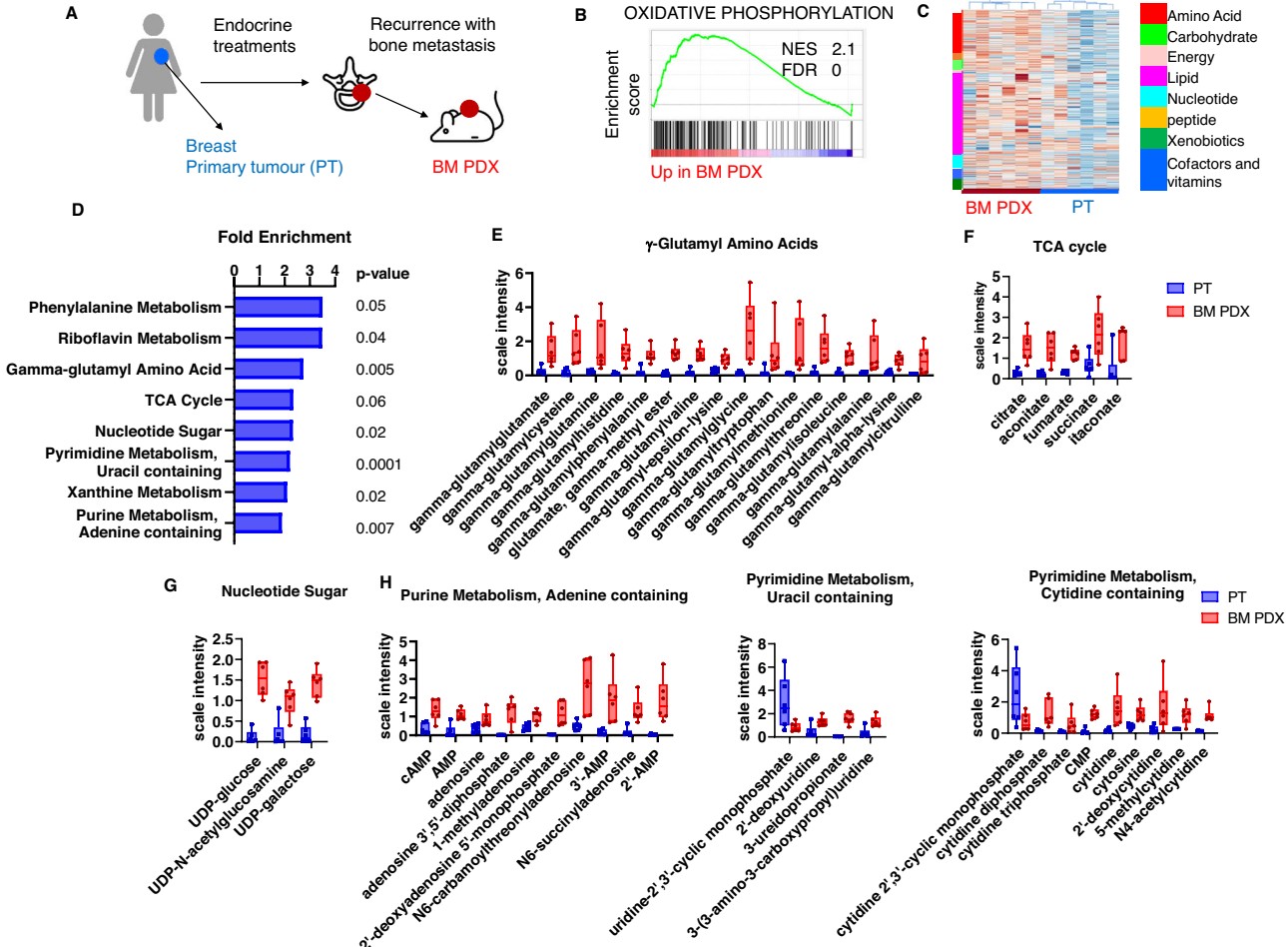

**Fig. 1 | Transcriptomic and metabolomics reprogramming in bone metastases-derived PDX as compared to patients' breast primary tumours. A** Biopsies of bone metastases (BM) from patients progressing after adjuvant treatments were engrafted in Swiss nude mice to generate PDX models. **B** Enrichment plot of oxidative phosphorylation hallmark identified in the Gene Set Enrichment Analysis (GSEA) of BM PDX ($n = 7$ PDX) as compared to patients' breast primary tumours (PT) ($n = 4$ patients), including four pairs of matched patients/PDX samples. NES: normalised enrichment score. FDR false discovery rate. **C** Hierarchical clustering analysis of metabolic profiles of primary breast tumours ($n = 6$ patients) and BM PDX samples ($n = 6$ PDX), including four matched patients/PDX samples. **D** Metabolite pathway enrichment analysis (MPEA) of differentially regulated metabolites between BM PDX and patients' PT. *P* value were calculated with a right-tailed Fisher's exact test. No correction for multiple testing. **E** Min/Max Whiskers plots showing levels of metabolites of γ-glutamyl amino acids in BM PDX (PDX) and patients' primary tumours (PT), **F** Min/Max Whiskers plots showing levels of metabolites of TCA cycle in BM PDX and PT), **G** Min/Max Whiskers plots showing levels of metabolites of nucleotide sugar in BM PDX and PT), **H** Min/Max Whiskers plots showing levels of metabolites of purine and pyrimidine metabolism in BM PDX and PT. In all Min/Max Whiskers plots, $n = 6$ (PT and BM PDX). PT patients' primary tumours, BM PDX bone metastases-derived PDX. Source data are provided as a Source Data file. *n* number of different patients' tumours or PDX models.

combination with palbociclib and fulvestrant in the HBCx-124 palboR25 PDX. Treatment was administered for 50 days and mice were followed up for tumour recurrence for 6 months. In both the monotherapy and combination setting, IACS treatment resulted in a marked antitumour activity with a median survival of 126 and 144 days, respectively, as compared to 30 days for the palbociclib + fulvestrant treated group (Fig. 2F) ($p < 0.0001$, log-rank (Mantel−Cox) test). Oxidative phosphorylation was also enriched in the second model of Palbociclib resistance, HBCx-134 palboR31 PDX (Fig. 2G, H). In this model, IACS treatment resulted in stable disease (Fig. 2I), a comparable effect to that observed in the parental HBCx-134 (Fig. 2A). This PDX model was also sensitive to the 5 mg/kg dose of IACS, while the 2.5 mg/kg dose was almost ineffective (Fig. S1C).

Collectively, these results indicate that targeting OXPHOS does not restore palbociclib sensitivity, that acquired resistance to palbociclib does not change response to OXPHOS inhibition and that OXPHOS targeting is an efficient treatment for palbociclib-resistant tumours.

To further analyse the metabolic reprogramming in a context of endocrine and palbociclib resistance, we used a panel of isogenic cell lines made resistant to long-term oestrogen deprivation (MCF7 LTED and T47D LTED) or to palbociclib (MCF7-PalboR), characterised by metabolic reprogramming[17–19]. These cell lines were subjected to escalating concentrations of IACS. MCF7-LTED and MCF7-PalboR cell lines were more sensitive, with an IC$_{50}$ of 4.7 and 0.6 nM, respectively, than the parental counterpart (IC$_{50}$: 2.5 μM) ($p < 0.0001$, two-way Anova) (Fig. 3A). Similarly, T47D LTED cells displayed an increased sensitivity to IACS as compared to the parental T47D (($p < 0.0001$, two-way Anova). Importantly, the higher sensitivity displayed by the resistant cells was paralleled by, the higher ratio of oxygen consumption rate (OCR) and extracellular acidification rate (ECAR) in the resistant cells, a proxy of OXPHOS dependency[20] (Fig. 3B).

Finally, to evaluate whether targeting OXPHOS could be effective in metastatic BC showing primary resistance to CDK4/6 inhibitors, we established two PDX from BM of patients treated with first-line palbociclib plus letrozole (Fig. 3C). Both patients showed progressive

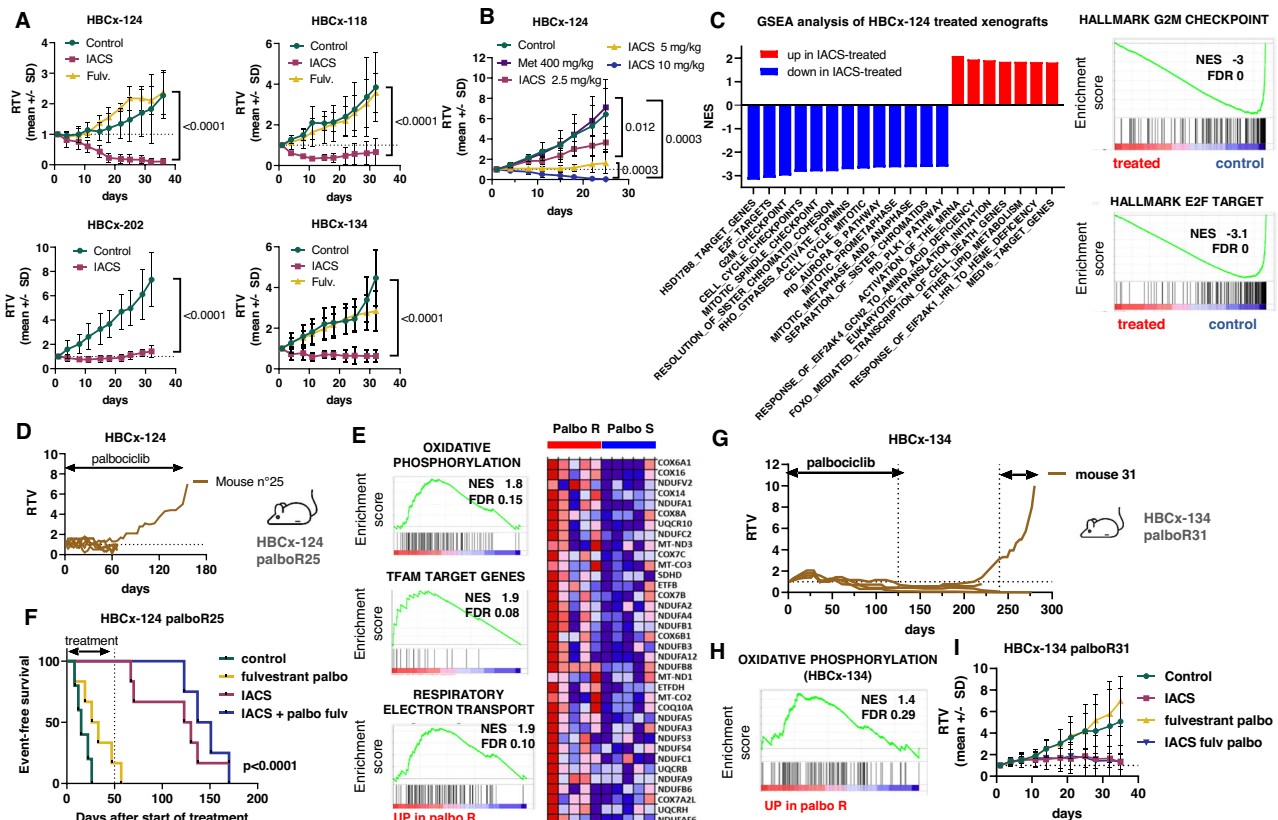

**Fig. 2 | Energy metabolism in BM PDX and OXPHOS targeting. A** In vivo response to IACS-010759 treatment in PDX of ER + BM. Mean ± SD. HBCx-124: $n = 3$ mice (control) and 4 mice (IACS and fulvestrant). HBCx-118: $n = 4$ mice/group. HBCx-202: $n = 6$ mice/group. HBCx-134 $n = 5$ mice (control and IACS) and $n = 8$ mice (fulvestrant). $P$ values were calculated with the Mann–Whitney test (two-tailed).
**B** Response of HBCx-124 to different doses of IACS-010759 and to metformin ($n = 7$ mice/group) $P$ values were calculated with the Mann–Whitney test (two-tailed).
**C** GSEA analysis of HBCx-124 xenografts after 1 week of IACS treatment.
**D** Establishment of the palbociclib-resistant HBCx-124 palboR25 PDX model.
**E** Enrichment plots of oxidative phosphorylation, TFAM target genes and respiratory electron transport hallmarks from the GSEA enrichment analysis of HBCx-124 palboR25 PDX as compared to the parental palbo responder (palbo S) HBCx-124.

NES normalised enrichment score. FDR false discovery rate. Oxidative phosphorylation heatmap. **F** Kaplan–Meier survival analysis of HBCx-124 palboR25 PDX treated by fulvestrant and palbociclib, IACS-010759 and the combination of palbociclib + fulvestrant and IACS-010759 $n = 6$ mice (control), $n = 7$ mice (fulv palbo and IACS), $n = 5$ mice (IACS + palbo fulv). Log-rank (Mantel–Cox) test.
**G** Establishment of the palbociclib-resistant HBCx-134 palboR31 PDX model.
**H** Enrichment plots of oxidative phosphorylation, NES normalised enrichment score, FDR false discovery rate. Oxidative phosphorylation heatmap. **I** Response to IACS, fulvestrant and palbo and IACS + fulvestrant and palbo in the HBCx-134 palbo31 PDX. $n = 4$ mice (control), $n = 5$ mice (IACS, palbo fulv), $n = 6$ mice (IACS + palbo fulv). Mean ± SD. RTV relative tumour volume. Source data are provided as a Source Data file. $n$ number of mice.

disease to palbociclib and letrozole. In the HBCx-180 PDX, IACS treatment as monotherapy arrested tumour growth, while the combination of IACS and palbo+ fulvestrant resulted in tumour shrinkage (Fig. 3D). In the HBCx-227, IACS treatment inhibited tumour growth without arresting tumour growth and the combination of IACS and palbo+ fulvestrant induced tumour regression.

Overall, on the 14 PDX used in the current study, that included 9 ER + BM-derived PDX, response to IACS treatment was observed in 6 ER + BM PDX models. An oncoplot in Fig. 4A summarises the PDX origin, the ER IHC status and the genomic alterations of the 14 PDX, including gene amplifications and pathogenic mutations. Interestingly, the PDX with the highest response to IACS (HBCx-124) (Fig. 2B) harbours an amplification of *MYC*, a regulator of biosynthetic and metabolic pathways, including OXPHOS[21,22]. Four PDX in the responder group have a mutation in *PIK3CA* or *AKT1* gene, whereas only one shows the same mutation pattern in the resistant group ($p = 0.036$, Chi-square test).

To gain insight into potential biological mechanisms driving response or resistance to IACS treatment, we performed an RNAseq analysis of the IACS responder and IACS-resistant untreated ER + PDX. The top enriched gene sets identified by GSEA analysis are shown in Fig. 4B and summarised in Table S2. Interferon response, oestrogen

response, PI3K/AKT/mTOR and OXPHOS hallmarks were significantly associated with IACS response, while proliferation hallmarks (E2F targets and G2M checkpoint) and EMT were positively associated with IACS resistance (Table S2 and Fig. 4C).

## Metabolic changes associated with response to the OXPHOS inhibitor IACS-010759

To identify metabolic changes associated with response or resistance to IACS-010759, we performed a global metabolomic analysis of untreated and treated xenografts from an IACS responder (HBCx-124) and an IACS resistant PDX (HBCx-137). The heatmaps of the top differentially expressed metabolites are shown in Fig. 5A. Statistical analysis revealed 199 and 163 differentially expressed metabolites between treated and control xenografts for the HBCx-124 and HBCx-137 PDX, respectively (Supplementary Data 2). The levels of 549 metabolites were significantly different between the HBCx-124 and HBCx-137 at baseline. Metabolite set enrichment analyses (MSEA) is shown in Fig. S2A: fatty acids (FA) metabolism, and branched-chain amino acids (Leucine, Isoleucine and Valine) were altered after IACS treatment in both PDX. The levels of several medium chain, long chain, monounsaturated, and polyunsaturated fatty acylcarnitines, which can serve as indicators

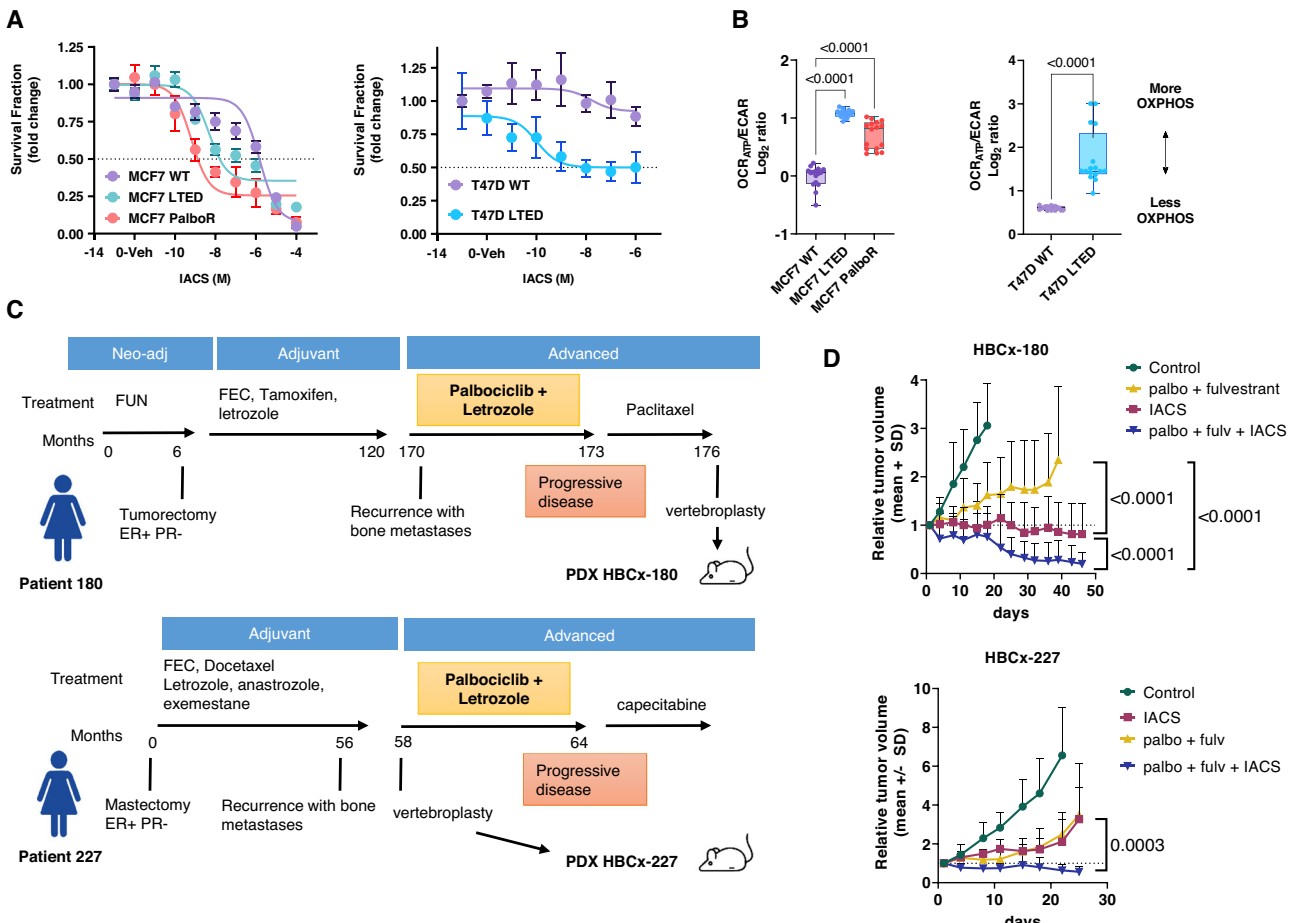

**Fig. 3 | OXPHOS targeting in cell lines and PDX models of palbociclib-resistant patients.** **A** Effect of IACS-010759 on in vitro models of ER+ breast cancer sensitive and resistant cells. Relative viability of MCF7 WT/LTED/ PalboR and of T47D WT/ LTED derivatives cells treated for 72 h with IACS (10⁻¹¹ to 10⁻⁴ M). Data represent mean survival fraction ±SEM relative to untreated cells (n = 3 independent experiments). Statistical analysis of MCF7 WT/LTED/PalboR was performed with the Two-way ANOVA test (Bonferroni corrected). Statistical analysis of T47D WT/LTED was performed with the unpaired multiple *t*-test. **B** Sensitive and resistant cells were subjected to Seahorse XFe96 Mito Stress Test and the ATP-dependent oxygen consumption rate (OCRATP)/extracellular acidification rate (ECAR) ratio was measured in real-time as an index of OXPHOS dependency. OCR and ECAR readings were determined for ten technical replicates from at least three biological replicates. Box and whisker plots with Min and Max values are represented. One-way ANOVA; Dunnett's corrected. **C** Clinical history of patients corresponding to PDX HBCx-180 and HBCx-227. **D** In vivo response to fulvestrant and palbociclib, IACS-010759 and the combination of palbociclib + fulvestrant and IACS-010759 in HBCx-180 and HBCx-227 (mean ± SD). HBCx-180: n = 8 mice (control, IACS, palbo fulv + IACS) and n = 7 mice (palbo fulv). HBCx-227, n = 9 mice (control), n = 7 mice (IACS), n = 8 mice (palbo fulv), n = 5 mice (palbo fulv. + IACS). Mann–Whitney test, Two-tailed. RTV relative tumour volume. Source data are provided as a Source Data file. *n* number of mice.

for changes in β-oxidation, were significantly reduced in response to IACS-010759 treatment in both PDX and were increased in the control samples of the responder tumour (HBCx-124) as compared to the resistant PDX (HBCx-137) (Supplementary Data 2). During FA β-oxidation, FA are catabolized to provide acetyl-CoA for energy generation via the TCA cycle (Fig. S2B). Long-chain FA are conjugated to carnitine for transport into the mitochondria through the carnitine shuttle. In the mitochondria, these fatty acylcarnitines are progressively shortened by two carbons through rounds of β-oxidation, generating NADH, FADH2, and acetyl-CoA. The levels of branched-chain amino acids (BCAA) (leucine, isoleucine and valine) were increased after IACS treatment in both PDX (Fig. S2C). Notably, the carbon skeleton of these amino acids can be metabolised into acetyl-coA and succinyl-CoA, and replenish the TCA cycle, thus contributing to ATP production (Fig. S2B).

The levels of 3-hydroxy FA and 3-hydroxy acylcarnitines were enhanced by IACS-010759 treatment in both responder and resistant models and their basal levels were significantly higher in the responding HBCx-124 PDX when compared to the resistant HBCx-137 (Supplementary Data 2).

Collectively, these results show that the responder tumour has higher basal levels of FA, and that OXPHOS inhibition impairs FA β-oxidation in both responder and resistant PDX.

Glutathione, which exists in either reduced (GSH) or oxidised (GSSG) state, is one of the primary anti-oxidant systems utilised to maintain redox homeostasis and counteract oxidative damage[23]. In the treatment-responder PDX HBCx-124, the levels of glutathione and its precursors (S-adenosylmethionine (SAM), homocysteine, cystathionine) were significantly reduced as compared to the resistant HBCx-137 (Fig. 5B and Supplementary Data 2). The levels of some of these precursors are decreased in IACS-treated samples of the responder model but not in the resistant xenografts (Fig. 5C). This suggests that the IACS-responder PDX has reduced glutathione metabolism compared to the treatment-resistant PDX, regardless of IACS exposure. In support of this, ophthalmate, which serves as a good indicator of GSH utilisation, is also decreased in the HBCx-124 as compared to the HBCx-137 (Fig. 5B). Moreover, the levels of different metabolites of ascorbate metabolism, which is associated with glutathione pathway and is involved in the detoxification of reactive oxygen species (ROS), were also decreased in treated samples of the responder PDX (Fig. 5C). The

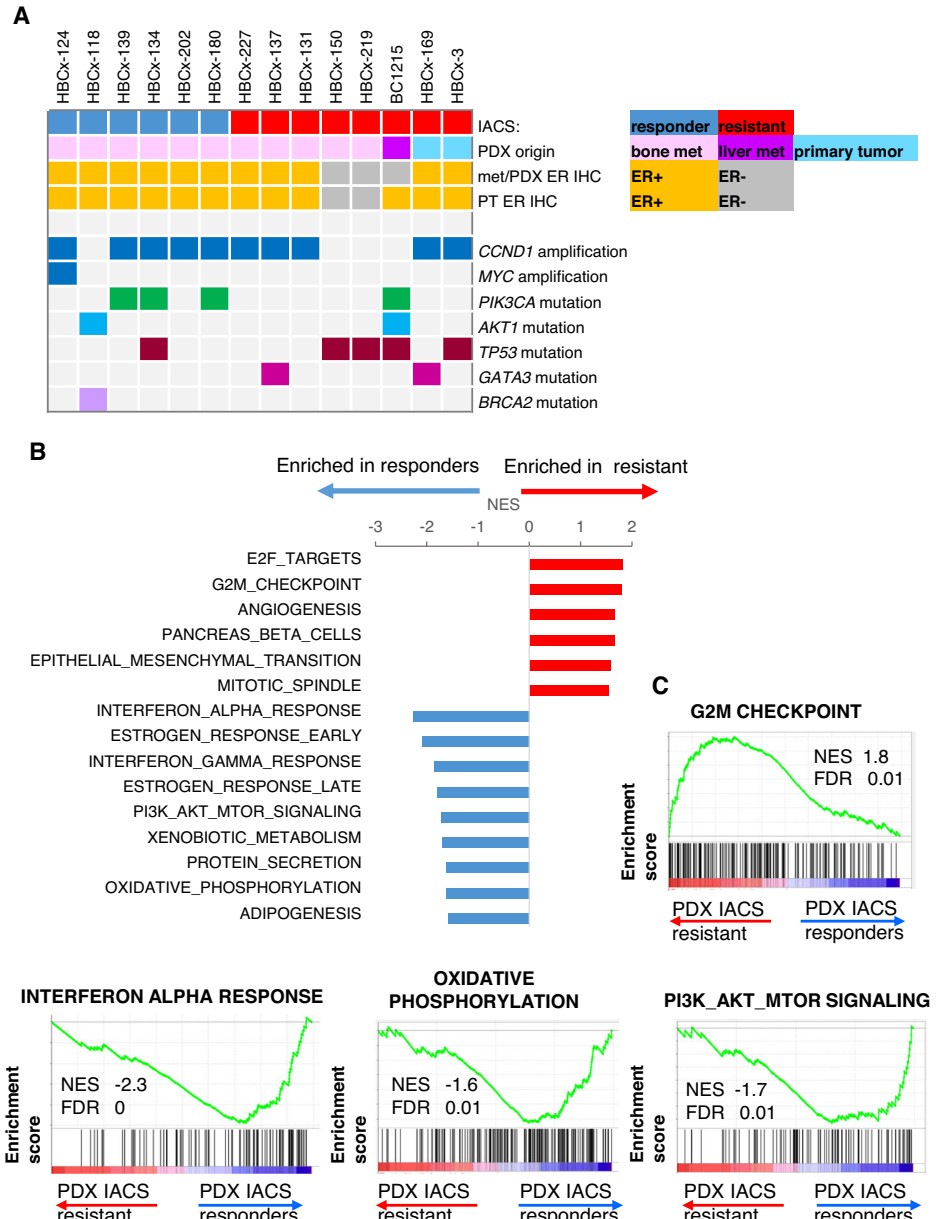

**Fig. 4 | Genomic analyses of PDX models. A** Oncoplot showing IACS response, the origin of PDX (bone metastasis, liver metastasis or breast primary tumour), the ER IHC status and the genomic alterations present in each PDX. **B** Top enriched hallmarks identified in the GSEA analysis of IACS responder and IACS resistant untreated ER + PDX models. Normalised enrichment scores (NES). **C** Enrichment plots of G2M checkpoint, interferon alpha response, OXPHOS and PI3K/AKT/mTOR hallmarks significantly associated with IACS response or resistance. NES normalised enrichment score, FDR false discovery rate adjusted *p* value.

metabolism of glutathione and ascorbate is represented in Fig. 5D. To determine the role of glutathione metabolism in response to IACS, we inhibited glutathione synthesis with the buthionine sulfoximine (BSO) ex vivo in tumour explants derived from the IACS resistant PDX (HBCx-137) and treated with different concentrations of IACS. Results showed a significant decrease in cell viability in cells treated by the combination of IACS and BSO (Fig. 5E). The relevance of ROS buffering in the effects exerted by IACS was investigated in vitro in the MCF7 LTED and PalboR cells, by monitoring H2O2-measuring fluorescent probes. The oxidation of 2′-7′ dichlorofluorescin (H2DCF) to 2′-7′dichloro-fluorescein (DCF) has been used quite extensively for the quantification of H2O2, although other ROS such as nitrate and hypochlorous acid can oxidise H2DCF[24]. Conversely, CellROX® green exhibits photostable fluorescence upon ROS-induced oxidation with subsequent binding to DNA, therefore localising its presence to the nucleus or mitochondria. We found that in both assays IACS treatment significantly enhanced ROS levels in the IACS-responder resistant cells (LTED and PalboR), hence suggesting a major imbalance in redox homeostasis when OXPHOS is inhibited (Fig. 5F).

PDX showed decreased levels of succinate after IACS treatment (Fig. 6A), suggesting potential changes in the TCA cycle. Moreover, the responder PDX (HBCx-124) exhibited higher levels of different TCA-related metabolites (citrate, aconitate and itaconiate) as compared to the resistant model (HBCx-137), in both untreated and treated samples that might be due to a higher TCA exploitation. The levels of aspartate, whose synthesis is supported by electron acceptors provided by mitochondrial respiration[25], were strongly reduced in both responder and resistant PDX after IACS treatment (Fig. 6B). Moreover, the basal levels of aspartate were higher in the responder model, which also showed increased levels of other intermediates of aspartate metabolism (Fig. 6B and Supplementary Data 2).

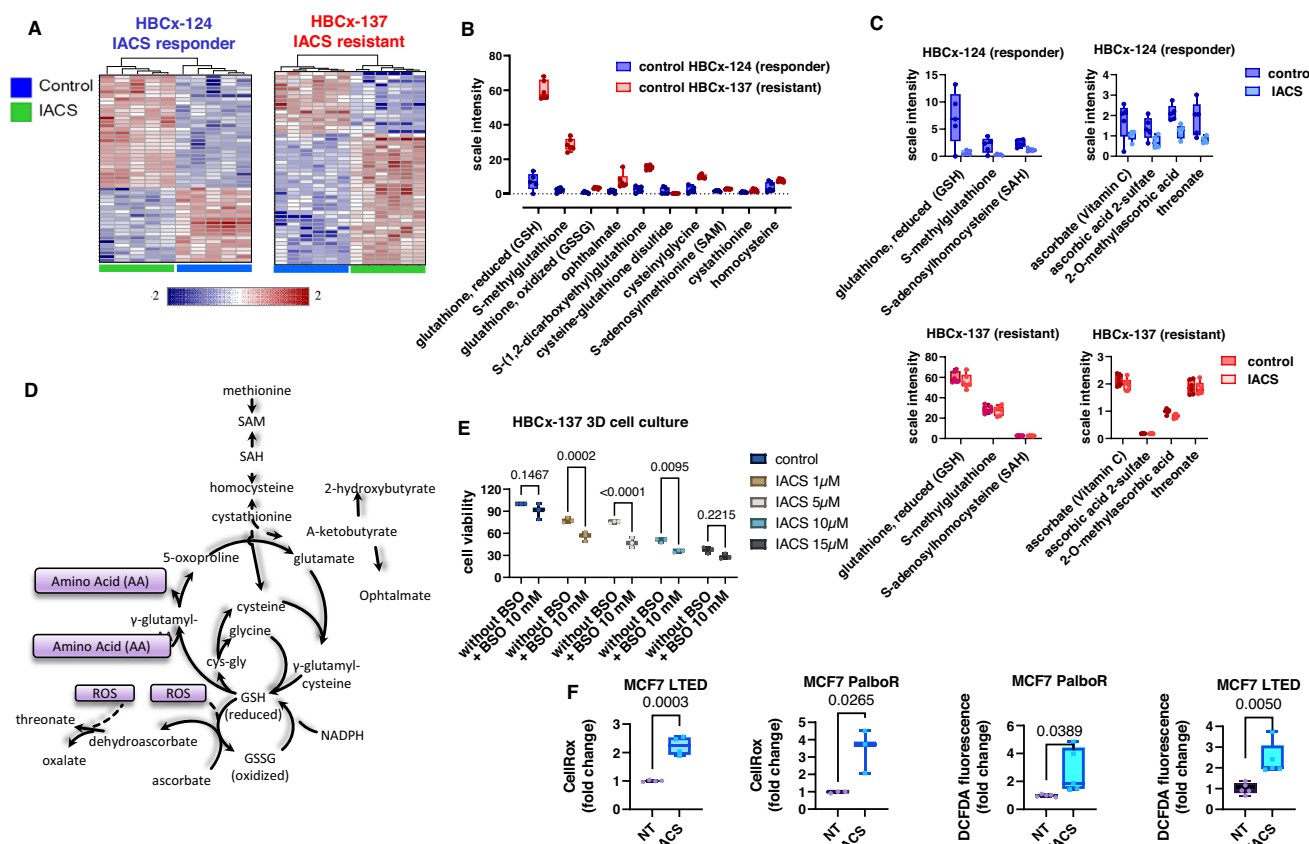

**Fig. 5 | Pharmacodynamic metabolomic analysis of HBCx-124 (IACS-responder) and HBCx-137 (IACS-resistant) PDX. A** Clustering results shown as heatmaps of the top differentially expressed metabolites in control and IACS-treated xenografts of HBCx-124 (n = 5 mice/group) and HBCx-137 PDX (n = 6 mice/group). Distance measure using Euclidean, and clustering algorithm using ward. **B** Levels of different metabolites of glutathione metabolism in control xenografts of HBCx-124 (n = 5 mice/group) and HBCx-137 PDX (n = 6 mice/group). Min/Max whisker plots with individual values. P and q values are shown in supplementary data 2. **C** Levels of different metabolites of glutathione and ascorbate metabolism in control and treated xenografts of HBCx-124 (n = 5 mice/group) and HBCx-137 PDX (n = 6 mice/ group). Min/Max whisker plots with individual values. P and q values are shown in Supplementary Data 2. n number of mice. **D** Methionine, cysteine, glutathione and ascorbate metabolism. **E** Cell viability of HBCx-137 derived 3D cell culture exposed to different concentrations of IACS with and without BSO (10 mM). (n = 3 technical replicates). Two-way ANOVA test, Šídák's multiple comparisons test. **F** MCF7-LTED and PalboR cells were treated with 1 µM IACS for 72 h and ROS levels were measured using CellROX and DCFDA staining. CellROX: n = 4 (MCF7 LTED), n = 3 (MCF7 palboR), DCFDA: n = 5 (MCF7 LTED and palboR). Unpaired Student t-tests (two-tailed). n number of technical replicates. Source data are provided as a Source Data file.

It was suggested from previous studies that OXPHOS inhibition is more effective in tumours that are not able to compensate with enhanced glycolytic metabolism to maintain ATP levels[15]. In our study, however, levels of glycolysis intermediates were similar in HBCx-124 and HBCx-137 samples, and production of lactate was significantly increased after IACS treatment in both PDX, suggesting increased glycolytic activity in post-treatment samples (Fig. 6C). By contrast, levels of different metabolites of the pentose phosphate pathway and related metabolism, which play an important role in the regulation of cell growth and survival, were significantly higher in the HBCx-124 PDX and were decreased exclusively in post-treatment samples of the responder PDX (Fig. 6D), while in the resistant HBCx-137 model, IACS treatment did not impair pentose metabolism. Moreover, maltotriose and maltotetraose metabolites, sugars associated with glycogen breakdown (glycogenolysis), were reduced in post-treatment samples of the IACS-responder as compared to the resistant PDX (Fig. 6E), suggesting that the resistant PDX may be able to initiate or maintain glycogenolysis in response to IACS treatment.

In summary, the IACS responder PDX is characterised by increased levels of mitochondrial and pentose phosphate pathway-related metabolites and decreased levels of glutathione at the basal level as compared to the resistant PDX, while the pharmacodynamics analysis identified inhibition of glycogen, pentose phosphate and glutathione metabolites in IACS-treated xenografts of the responder PDX. Up-regulation of glycolysis and BCCA catabolism were observed in treated tumours of both resistant and responder PDX (Fig. 6F).

## High expression of OXPHOS-related genes is associated with worse survival in breast cancer patients

Based on the OXPHOS signature enrichment in the metastases-derived samples, we hypothesised that the expression of genes associated with mitochondrial metabolism could be correlated with disease progression in BC patients. Therefore, we analysed the expression of 6 genes from the OXPHOS signature (AIFM1, NDUFV1, NDUFAB1, NDUFA7, NDUFS6 and MRPS12) among the most enriched in metastatic samples, by RT-PCR in specimens of 503 breast tumours patients with a follow-up of 20 years. The clinical characteristics of the 503 patients are summarized in Table S3. First, we assessed the correlation between the six genes in the whole cohort by performing a Pearson correlation analysis. NDUFS6 and MRSP12 were the most correlated genes, as shown in the correlation matrix (Fig. S3A). Then, we analyzed the expression of these 2 genes in the different BC subtypes (Fig. 7A) and as in relationship with the different biological and clinical parameters. The expression levels of both genes are higher in the ERBB2+ (HER2+) and TNBC subtypes as compared to ER+ tumours (RH+). Expression of MRPS12 and NDUFS6 were significantly associated with the histological

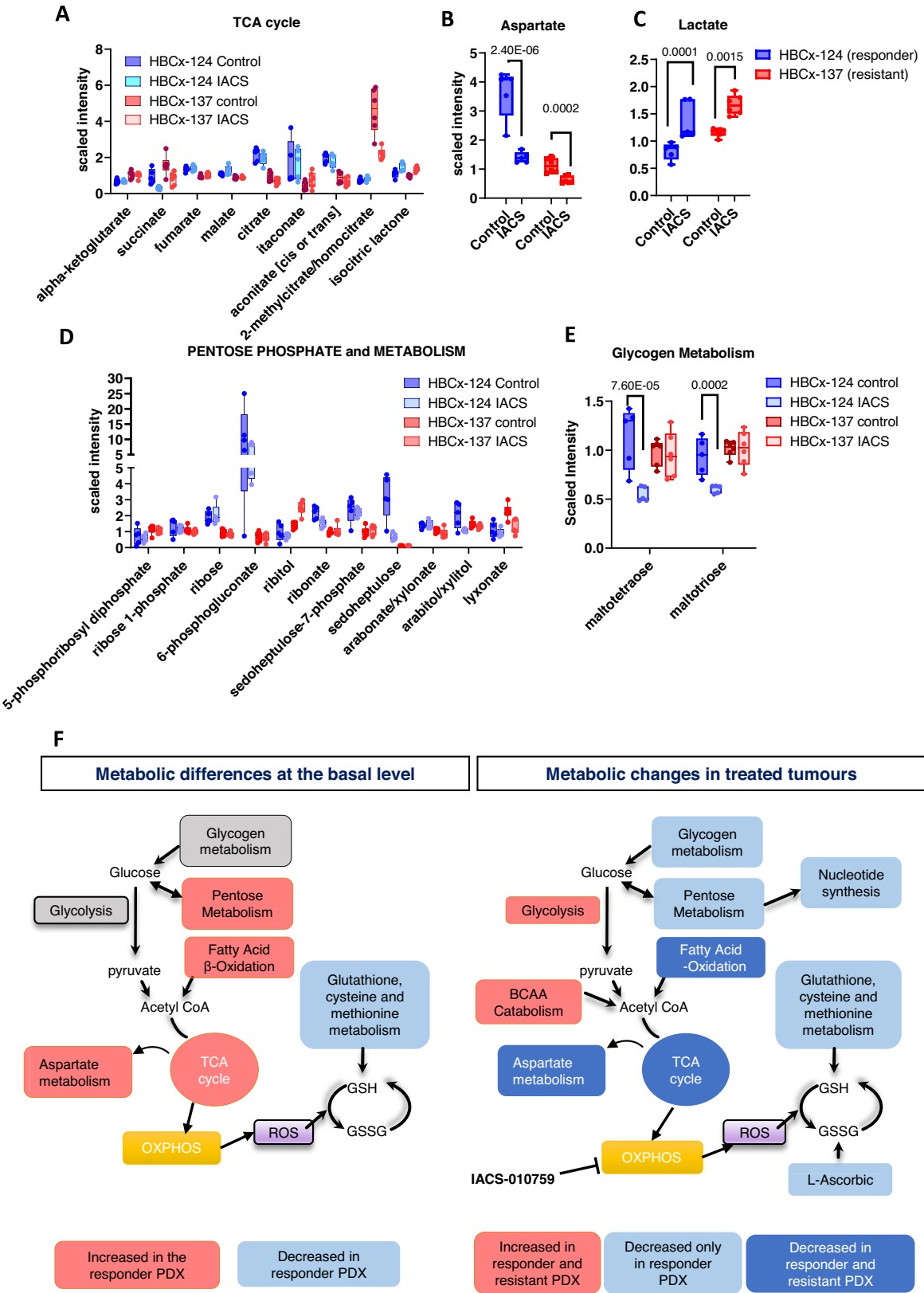

**Fig. 6 | Effect of OXPHOS inhibition on energy metabolism. A** Min/Max whisker plots showing the levels of TCA cycle metabolites in HBCx-124 ($n = 5$ mice/group) and HBCx-137 PDX ($n = 6$ mice/group) control and IACS-treated xenografts. (Mean ± SD). $P$ and $q$ values are shown in Supplementary Data 2. Level of aspartate (**B**) and lactate (**C**) in HBCx-124 ($n = 5$ mice/group) and HBCx-137 PDX ($n = 6$ mice/group) control and IACS-treated xenografts. (Mean ± SD). Min/Max whisker plots. (Two-way ANOVA, FDR corrected). **D** Metabolites levels of the pentose phosphate pathway metabolism ($P$ and $q$ values are shown in Supplementary Data 2.) in HBCx-

124 ($n = 5$ mice/group) and HBCx-137 PDX ($n = 6$ mice/group) control and IACS-treated xenografts. (Mean ± SD). Min/Max whisker plots. **E** Glycogen metabolism in HBCx-124 ($n = 5$ mice/group) and HBCx-137 PDX ($n = 6$ mice/group) control and IACS-treated xenografts. (Mean ± SD). Min/Max whisker plots. (Two-way ANOVA, FDR corrected). **F** Overview of metabolic differences between HBCx-124 and HBCx-137 PDX and of metabolic changes in IACS-treated tumours. Source data are provided as a Source Data file. $n$ number of mice.

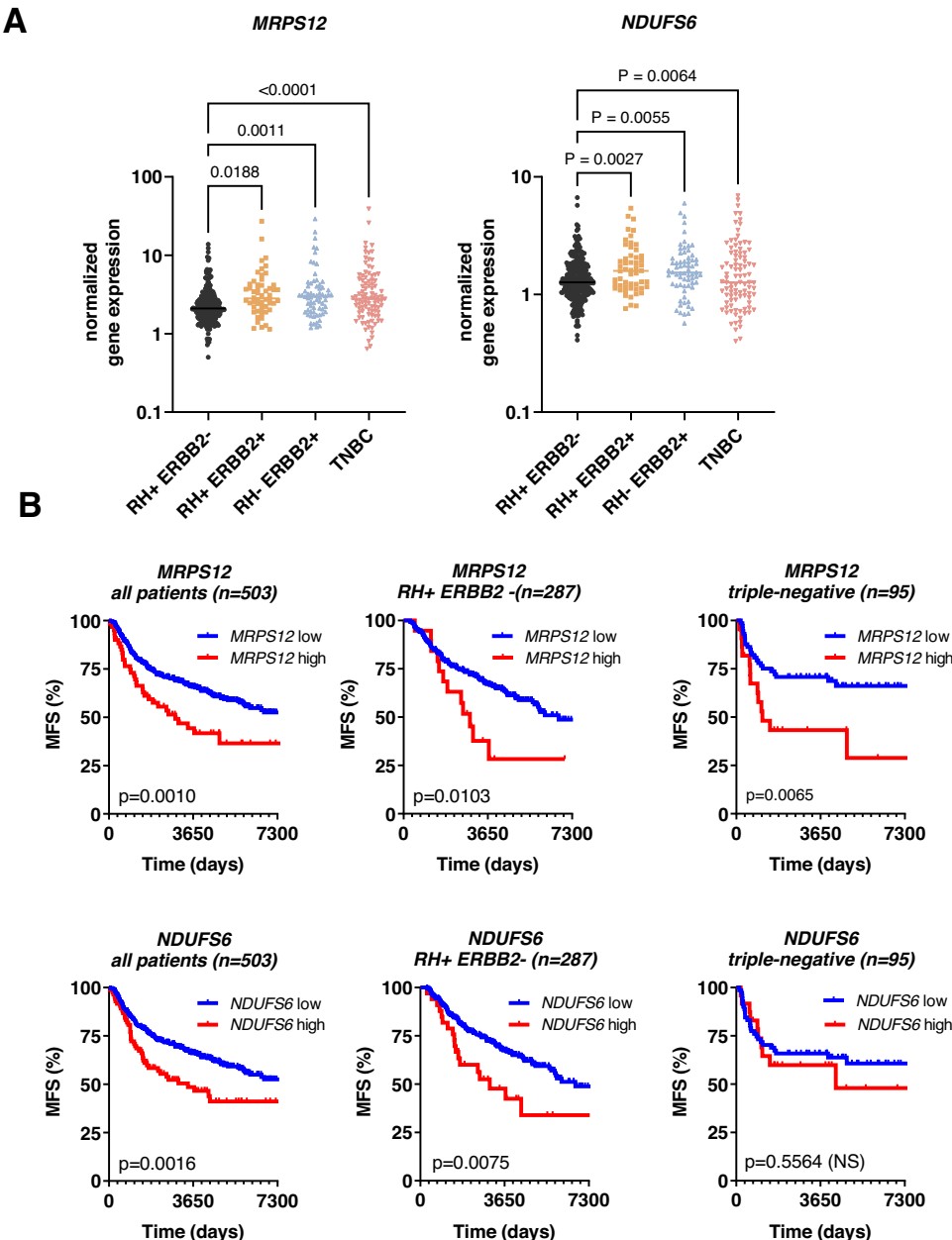

**Fig. 7 | RT-PCR analysis of OXPHOS-related genes in a cohort of 503 breast tumours. A** Scatter plot showing *NDUFS6* and *MRPS12* expression in different sub-groups of breast cancer: RH + ERBB2- (*n* = 287 patients), RH + ERBB2+ (*n* = 54 patients), RH- ERBB2+ (*n* = 67 patients) and TNBC (*n* = 95 patients). Scatter dot plots, median and individual values are shown. *P* values: one-way Anova, Turkey corrected. RH hormone receptor, ERBB2 Receptor tyrosine-protein kinase erbB-2, TNBC triple-negative breast cancer. Source data are provided as a Source Data file. **B** Metastasis-free survival (MFS) of breast cancer patients according to *MRPS12* and *NDUFS6* low and high expression (optimal cut-off). *P* values were calculated with the log-rank (Mantel−Cox) test. *n* number of patients.

grade, the ER and PR status, and the development of metastasis (Table S4). Moreover, we found a strong correlation between high *MRPS12* and *NDUFS6* expression and expression of the proliferation marker Ki67 ($p < 0.0001$). Finally, we analysed the prognostic significance of *MRPS12* and *NDUFS6* expression pattern: results in the global population (503 patients) showed that metastasis-free survival (MFS) of patients with high *MRPS12 or NDUFS6* expressing tumours (10-year MFS of 41 and 46%, respectively) was shorter than that of patients with low *MRPS12 or NDFUS6* expressing tumours (10-year MFS of 66% for both genes;) ($P = 0.001$ for *MRPS12* and $p = 0.0016$ for *NDFUS6*, Fig. 7B). The survival analysis in the different patients' sub-groups showed that high expression of *MRPS12* was associated to a worse prognosis in the RH+ and triple-negative BC patients, while *NDFUS6* was associated with a worse prognosis only in RH + BC

patients. No significant correlation with MFS was found in the ERBB2-positive tumours (Fig. S3B).

## Discussion

In this study, we showed activation of the OXPHOS signature in BM-derived PDX as compared to matched breast PT. Different groups identified OXPHOS metabolic reprogramming in preclinical models of metastatic BC. Dupuy et al. revealed activation of OXPHOS in bone and lung metastases of BC, in contrast to liver metastases that were more dependent on glycolysis[26]. Hu et al. identified an OXPHOS-dependent metabolic reprogramming in BC cells under osteogeneic differentiation[27]. Moreover, LeBleu et al. demonstrated that migratory/invasive capacities of murine mammary cancer cells specifically favour mitochondrial respiration and increased ATP production[28]. The

functional importance of OXPHOS in BC metastases was also demonstrated through single-cell RNAseq analysis of micrometastases in BC PDX[12]. As we could not analyse patients' biopsies of BM, we cannot exclude that activation of OXPHOS in PDX models could be partially driven by the tumour engraftment in mice. However, the finding that not all the BM PDX rely on OXPHOS, as some were resistant to the OXPHOS inhibitor, suggests a tumour-dependent activation of OXPHOS. This is supported by the finding that in patients' samples, OXPHOS is one of the up-regulated pathways in gene expression data of BM as compared to primary BC[8]. In addition, RNAseq analysis of matched breast tumours and brain metastases showed enrichment of OXPHOS[29,30].

As compared to breast PT, BM-derived PDX display a profound metabolic reprogramming. Levels of several gamma-glutamyl amino acids and other metabolites from the glutathione and cysteine pathways were increased in BM PDX, suggesting an increased resistance to oxidative stress. Increased levels of GSH may play a role in chemoresistance due to the increased capacity of these cells to reduce oxidative stress[23].

The levels of several TCA cycle intermediates are also significantly elevated in PDX. The TCA cycle is closely coordinated with OXPHOS, as it produces FADH2 and NADH, that are utilised in the mitochondrial respiratory chain for electron transfer[14]. Increased metabolites of the pentose phosphate pathway and glycogen metabolism were also observed. Glycogen serves as a storage form of glucose and its metabolism has been demonstrated to be altered in many tumour types[31], and in metastases[32]. While it is unclear if the increased levels of these metabolites reflect increased glycogen synthesis or breakdown, previous studies have shown that metastatic tumours contain higher glycogen levels than primary tumours[33]. High levels of hexosamine pathway metabolites suggest increased glycosylation in BM PDX samples. Cancer-associated carbohydrate structures play key roles in cancer progression and glycosylation changes are a universal feature of malignant transformation and tumour progression[34–36].

Targeting BM PDX by the complex I inhibitor IACS-010759, significantly reduced tumour growth in the majority of ER + BM PDX, including multiple endocrine-resistant models and PDX established from patients progressing on palbociclib. Conversely, it was found ineffective in triple-negative BM PDX and PT-derived PDX. However, the number of models tested in these groups of tumours was not sufficient to conclude a lack of activity. A recent publication reported the activity of IACS-010759 in some TNBC PDX models[37]. Metformin, an antidiabetic drug that has anti-proliferative activities through multiple mechanisms dependent and independent of complex I inhibition, did not impair tumour growth when administered at a high dose. This might be because metformin is less potent and specific than IACS-010759[38], in some cellular models, it induces apoptosis at concentrations at which complex I is not fully inhibited[38], suggesting that its previously reported antitumour effects may also result from the inhibition of other targets.

Interestingly, IACS-010759 treatment resulted in a striking antitumour activity also in PDX models with acquired resistance to the CDK4/6 inhibitor palbociclib that exhibited enrichment of OXPHOS and TFAM target genes when compared to the parental CDK4/6 responder PDX. These results demonstrate that OXPHOS is a target in treatment-resistant metastatic BC, including tumours with primary and secondary resistance to CDK4/6 inhibitors.

Mutations of *PIK3CA* gene, which are present in approximately 40% of ER + BC[39] *or AKT1* genes, were significantly associated with IACS response in our cohort of PDX. Moreover, we found that the most responder model carried a *MYC* amplification, which occurs in approximately 20% of BC. Our results suggest that mutations of *PIK3CA/AKT1* genes and *MYC* amplifications could be used as potential biomarkers to identify patients who are likely to derive benefit from OXPHOS inhibitors.

The GSEA analysis of IACS responder and IACS resistant models confirmed the PI3K/AKT/mTOR hallmark as significantly associated with response, together with OXPHOS and the interferon response. Interestingly, mTOR signalling has been shown to promote mitochondrial biogenesis and activate OXPHOS[40]. Moreover, Evans et al. found a decreased level of P-AKT in IACS-treated PDX models, further suggesting a role of this signalling pathway in response to OXPHOS inhibitor[37].

In the comparative metabolomics analysis, the responder PDX showed higher levels of TCA cycle, FA beta-oxidation, and aspartate metabolism at the basal level, indicating increased mitochondrial bioenergetics (Fig. 6F). Conversely, the basal levels of glutathione, cysteine and methionine metabolism were strongly reduced as compared to the resistant PDX, suggesting a decreased capability to counteract oxidative stress. In addition, different metabolites of the glutathione, methionine and cysteine pathways were decreased in treated samples of the responder PDX, but not in the resistant. Similar results were reported in PDX models of T-cell lymphoblastic leukaemia that exhibited decreased levels of glutathione after IACS treatment[41]. Interestingly, glutathione metabolism was also inhibited upon metformin (another OXPHOS inhibitor) treatment in preclinical models of ovarian cancer and in metformin-exceptional responder patients[42], further supporting its role in determining the response to OXPHOS inhibitors.

Short-chain acylcarnitines, involved in mitochondrial metabolism[43], were decreased in treated samples of both responder and resistant PDX, consistently with previous studies on the metabolic effects of metformin treatment in ovarian cancer patients[42]. Similarly, decreased levels of several medium chain, long-chain, mono-unsaturated, and polyunsaturated fatty acylcarnitines, in treated samples of both responsive and resistant PDX indicate decreased β-oxidation. This supports previous findings that impairment of OXPHOS can lead to a reduction in mitochondrial β-oxidation[44]. By contrast, treatment with IACS-010759 enhanced the levels of 3-hydroxy FA and 3-hydroxy acylcarnitines in both PDX models. Interestingly, the accumulation of long-chain 3-hydroxy FA, such as 3-hydroxydecanoate have also been demonstrated to behave as uncouplers of oxidative phosphorylation[45]. Endogenous uncoupling proteins have been shown to activate the electron transport chain in response to increased levels of ROS or FA[46]; therefore the increased production of 3-hydroxy FA may be a homeostatic response to increase electron transport chain activity in order to counteract the inhibitory effects of IACS-010759.

Compensatory glycolysis, indicated by increased levels of lactate in treated samples, was observed in both resistant and responder models (Fig. 6F), and therefore is not discriminating response and resistance to IACS-010759 in our PDX models, as it was previously demonstrated for other cancer types[15]. By contrast, different metabolites of the glycogen and pentose phosphate pathways were decreased exclusively in treated xenografts of the responder PDX, suggesting a reduced capacity to maintain glycogen breakdown as a complementary source of energy and potential impairment of nucleotide synthesis.

Our study has some limitations. First, the pharmacodynamics analysis was limited to 2 PDX, therefore, we cannot exclude the existence of additional metabolic adaptations in response to OXPHOS inhibition, either in the responder or in the resistant models. Second, we cannot exclude the existence of off-target effects of the OXPHOS inhibitor dosed at 10 mg/kg in the responder models.

Finally, we provide evidence that high expression of the OXPHOS-related genes *NDUDS6* and *MRPS12* in breast tumours is associated to poor prognosis in a large cohort of BC patients. Mitochondrial genes have been found to predict recurrence and metastasis in an in silico analysis performed in a cohort of 145 ER-positive BC patients[47]. Recently, a RNAseq analysis of patients' breast tumours identified

OXPHOS and mitochondrial ATP synthesis as significantly associated with acquired resistance to endocrine therapies[48]. Interestingly, oxidative phosphorylation was also found to be associated with palbociclib resistance in the Paloma 2 and 3 clinical trials[49].

Despite the impressive response to OXPHOS inhibitors that we and others reported in preclinical models of different cancer types, clinical development of agents targeting metabolism is challenging. A recent phase I trial on IACS-010759 in patients with relapsed/refractory acute myeloid leukaemia (AML) or advanced solid tumours reported dose-limiting toxicities, including elevated blood lactate and neurotoxicity[50]. Peripheral neuropathy was minimised in preclinical models with the coadministration of a histone deacetylase 6 inhibitor, suggesting that combination approaches may help mitigate the neurotoxic effects associated with OXPHOS inhibitors[50,51]. Other potential strategies to reduce complex I inhibitors-related toxicities may include the development of antibody-drug conjugates (ADC) to achieve a greater selectivity towards cancer cells or the identification of pre-existing conditions that may predispose patients to neural damage, such as allogeneic stem cell implantation for AML patients or concomitant diagnosis of diabetes mellitus[50].

In conclusion, our study provides evidence that OXPHOS is a metabolic vulnerability in preclinical models of ER+ metastatic BC with resistance to endocrine treatments and CDK4/6 inhibitors.

## Methods

### Ethics statement
Human breast tumour fragments were obtained with patients' informed consent (Institute Curie).

In vivo experimental procedures were approved by the Institutional Animal Care and French Committee (project authorisation no. 02163.02) and were performed according to institutional regulations.

### Patient-derived xenografts (PDX) establishment
PDX models of ER+ metastatic BC were obtained by engrafting biopsies from spinal bone metastases of ER-positive BC female patients progressing under ET and treated with vertebroplasty to restore biomechanical vertebral properties (stabilisation) and reduce back pain, as described previously[13]. The protocol was approved by the Institut Curie Hospital committee (CRI: Comité de Revue Institutionnel). Bone metastasis biopsies were engrafted with informed consent from the patient into the interscapular fat pad of 8- to 12-week-old female Swiss nude mice (Charles River Laboratories), which were maintained under specific pathogen-free conditions. Their care and housing were in accordance with institutional guidelines and the rules of the French Ethics Committee: CEEA-IC (Comité d'Ethique en matière d'expérimentation animale de l'Institut Curie, National registration number: #118). The project authorisation no. is 02163.02. The housing facility was kept at 22 °C (±2 °C) with a relative humidity of 30–70%. The light/dark cycle was 12 h light/12 h dark. Mice were maintained on a standard diet (4RF25, Mucedola SRL, Italy) and were given free access to food and water.

### Targeted sequencing of PDX
Next generation targeted sequencing (NGS) of PDX tumour samples was performed with a 500-1000X coverage as described in detail elsewhere[13,52]. Briefly, NGS was performed on an Illumina HiSeq2500 sequencer and the genomic variants were annotated with COSMIC and 1000 genome databases[53]. Reads were aligned using the Burrows-Wheeler Aligner (BWA) software, allowing up to 4% of mismatches with the reference. Variants with low allelic frequency (<5%) or low coverage (<100x) were excluded from the analysis.

PDX HBCx-124, HBCx-134, HBCx-169, HBCx-180, HBCx-219, HBCx-202 and HBCx-227, were sequenced with a targeted NGS panel composed of 571 genes of interest in oncology[13]. PDX HBCx-3, HBCx-118, HBCx-139, HBCx-137, HBCx-131 and BC1215 were analyzed by targeted NGS of 95 genes, chosen among the most frequently mutated genes in breast cancer (>1%)[13,52].

NGS primers were selected based on their specificity on the human genome.

### In vivo efficacy studies
IACS-010759 and palbociclib were purchased from MedchemExpress and were administered orally 5 days per week at a dose of 10 and 75 mg/kg, respectively. Fulvestrant (Faslodex, AstraZeneca, Macclesfield, UK) was administered by intramuscular injection at 50 mg/kg once a week. Metformin (ARROW LAB) was administered orally at 400 mg/kg/day (BID).

For efficacy studies, tumour fragments were transplanted into female 8-week-old Swiss nude mice. Xenografts were randomly assigned to the different treatment groups when tumours reached a volume comprised between 100 and 200 mm³. Tumour size was measured with a manual calliper twice per week. Tumour volumes were calculated as $V = a \times b^2/2$, $a$ being the largest diameter, $b$ the smallest. The maximal tumour size/burden (1500 mm³) was not exceeded. Tumour volumes were then reported to the initial volume as relative tumour volume (RTV). Means (and SD) of RTV in the same treatment group were calculated, and growth curves were established as a function of time. For each tumour, the percent change in volume was calculated as (Vf – V0/V0)/100, V0 being the initial volume (at the beginning of treatment) and Vf the final volume (at the end of treatment). A decrease in tumour volume of at least 50% was classified as regression, an increase in tumour volume of at least 35% identified progressive disease and volume changes between +35% and −50% were considered as stable disease[54]. The statistical analysis of tumour growth inhibition was performed with the Mann−Whitney test or Dunn's multiple comparisons test. For the event-free survival analysis, an event in the solid was defined as a threefold tumour volume from the initial tumour volume. Event-free survival was defined as the time interval from the initiation of the study to the first event. The Kaplan−Meier survival was plotted with GraphPad software and the log-rank (Mantel−Cox) test was used for the statistical analysis. In vivo experiments were not replicated.

### Gene expression analysis
For gene expression microarray analyses, GeneChip Human 1.1 ST arrays were hybridised according to Affymetrix recommendations, using the Ambion WT Expression Kit protocol (Life Technologies) and Affymetrix labelling and hybridisation kits, as previously described in ref. 13. Gene set enrichment analysis (GSEA) software and MsigDB database[55] were used to identify overrepresented biological functions for differentially expressed genes between PDX and primary breast tumours.

### RNAseq of PDX samples
Total RNA was extracted from breast tumour xenografts samples by using miRNeasy Mini kit (#217004, Qiagen) according to the manufacturer's procedure, treated by RNase-Free DNase (#79254, Qiagen), quantified and quality-controlled using a 2100 Bioanalyzer (Agilent).

RNA sequencing libraries were prepared from 500 ng of total RNA using the Illumina TruSeq Stranded mRNA Library preparation kit, which allows to perform strand-specific sequencing. A first step of polyA selection using magnetic beads was done to address sequencing, specifically on polyadenylated transcripts. After fragmentation, cDNA synthesis was performed and resulting fragments were used for dA-tailing followed by ligation of the TruSeq indexed adaptors (Unique Dual Indexing strategy). PCR amplification was finally achieved, with 13 cycles, to generate the final cDNA libraries. After qPCR quantification using the KAPA library quantification kit (Roche), Sequencing was carried out on the NovaSeq 6000 instrument from Illumina based on a 2 × 100 cycle mode (paired-end reads,

100 bases) to obtain around 100 million clusters (200 million raw paired-end reads) per sample.

For sequenced PDX transcriptomes, reads were aligned to human genome reference (hg19) and annotated with the human GENCODE (v19) database using the STAR tool (v.2.7.5a)[56]. In order to filter mouse ARN contamination, alignment and annotation were also performed on mouse genome reference (mm10) and GENCODE (vM22) features, and both human and mouse results were used as input to XenofilteR R package[57]. Read counts per human GENCODE features were obtained by HTSeq-count[58]. DESeq2 was used to correct batch effect, get normalised data and make paired differential expressions between all conditions.

Gene set enrichment analysis (GSEA, v4.0.3) software and MsigDB database (v7.1)[55] were used to identify overrepresented biological functions for differentially expressed genes resulted from DESeq analysis/detected in the different comparisons.

### Global metabolomics analysis

Global metabolomic profiling of primary tumours and PDX samples was performed using the analytical DiscoveryHD4 platform by Metabolon[59].

Briefly, frozen samples were extracted with methanol to precipitate proteins and dissociate small molecules bound to protein or trapped in the precipitated protein matrix, followed by centrifugation to recover chemically diverse metabolites. The resulting extracts were analysed by ultrahigh performance liquid chromatography–tandem mass spectroscopy (UPLC-MS/MS) using a Waters ACQUITY ultraperformance liquid chromatography (UPLC) and a Thermo Scientific Q-Exactive high resolution/accurate mass spectrometer (MS) interfaced with a heated electrospray ionisation (HESI-II) source and Orbitrap mass analyzer operated at 35,000 mass resolution[60]. The UPLC-MS/MS platform included different chromatography methods and mass spectrometry ionisation modes to achieve broad coverage of compounds differing by physiochemical properties, as detailed previously[61].

Metabolites were identified by comparison of the ion features in the experimental samples to a reference library of chemical authenticated standards that included the molecular weight (m/z), the retention time and the MS/MS spectral data. Metabolomic data were extracted and analysed using an informatics system consisting of four major components, LIMS (Laboratory Information Management System), the data extraction and peak-identification software, data processing tools for QC and compound identification, and a collection of statistical, visualisation, and interpretation tools. The hardware and software foundations for these informatics components are the LAN backbone and database servers running Oracle 10.2.0.1 Enterprise Edition.

### Statistical analysis of metabolomic data

Standard statistical analyses were performed in Array Studio on log-transformed data. Welch's two-sample $t$-test and the two-way ANOVA test were used to analyse the data.

Both Welch's two-sample $t$-test and matched pairs $t$-test were performed for the patients' PT vs PDX dataset. The two-way ANOVA with FDR correction for multiple testing was used to analyse the different PDX and groups (IACS/control). Hierarchical clustering were used to analyze the data. For all analyses, following normalisation to a mass of the sample extracted, missing values, if any, were imputed with the observed minimum for that particular compound. The statistical analyses were performed on natural log-transformed data.

### Metabolite set enrichment and pathway analysis

Pathway enrichment analysis was performed using MyMetabolon pathway explorer (Metabolon, Inc). Enrichment values were calculated using the following equation:

Enrichment value = (k/m)/((n-k)/(N-m)), Where m = number of metabolites in the pathway; k = number of significant metabolites in the pathway; n = total number of significant metabolites; and N = total number of metabolites. $P$ values for each pathway are calculated using a right-tailed Fisher's exact test based on the hypergeometric distribution.

Normalised values of all metabolites detected are presented in Supplementary Data 3.

### Cells

MCF7 (ATCC HTB-22) and T47D (ATCC HTB-133) human breast cancer cells were obtained from ATCC and cultured in phenol red–free RPMI 1640 medium supplemented with 10% fetal bovine serum (FBS, Euroclone), 2 mmol/L glutamine and 1 nmol/L 17β-estradiol (E2, Sigma). The corresponding LTED derivatives were maintained in a sterol-deprived medium in phenol red–free RPMI 1640 medium containing 10% dextran charcoal-stripped FBS (Hyclone) and 2 mmol/L glutamine (DCC medium). The palbociclib-resistant derivatives (PalboR) were cultured in in DCC medium in the presence of 1 μM palbociclib (Pfizer, Pfizer Italia, Latina, Italy). Cells were short tandem repeat tested, amplified, stocked, routinely subjected to mycoplasma testing and once thawed were kept in culture for a maximum of 20 passages.

### Cell viability assay

Sensitive and resistant cells were seeded into 12-well plates in either standard conditions or experimental conditions in the presence of escalating doses ($10^{-11}$–$10^{-4}$ M) of IACS-010759. Three days post cell seeding, plates were stained with crystal violet (triphenylmethane dye 4-[(4-dimethylaminophenyl)-phenyl-methyl]-N,N-dymethyl-alanine; Sigma #548-62-9), dried overnight and the crystal violet within the adherent cells solubilized using 0.5 ml/well of 2% SDS. The absorbance at 595 nm was monitored using a microplate reader.

### Cell viability assay for PDX-derived 3D cell cultures

Ex vivo evaluation of cell viability in response to IACS-010759 exposure was performed in HBCx-137 derived cell cultures, following a protocol detailed elsewhere[62]. Briefly, fresh breast cancer PDX tissues were minced and further dissociated using a digestion medium. This was followed by further dissociation using trypsin (GIBCO), Dispase (StemCell Technologies, 7913) and DNase (Sigma, D4513). Cells were resuspended in MEGM (Lonza, CC-3150) and falcon® 40-μm cell strainers were used to remove the undigested tissue. Single-cell suspensions generated from tumours were plated in triplicates at 40,000 cells/well into 96-well plates. The effect of IACS treatment (0 to 15 μM– 48 h), with and without Buthionine sulphoximine (BSO) (Sigma, B2515) at 10 mM, on cell viability was determined by CellTiterGlo assay (Promega, G7571) and drug responses represented by the half-maximal inhibitory concentration (IC50) and the dose-response curve. The experiment was performed in two biological replicates. Cell viability was normalised to DMSO (vehicle) treated cells.

### Seahorse-based oxidative phenotyping

The oxygen consumption rates (OCR) and the extracellular acidification rates (ECAR) of cell lines were quantified using the Seahorse Extracellular Flux Analyzer (XFe96, Agilent). Cells were seeded in XFe96 cell culture plates with 1.5–$2 \times 10^4$ cells per well in 80 μL of culture medium and incubated at 37 °C (8–10 technical replicates). Twenty-four hours post-seeding medium was replaced with 180 μL XF base medium supplemented with 11 mM glucose, 2 mM glutamine and 1 mM sodium pyruvate. Cells were incubated for 1 h at 37 °C in atmospheric $CO_2$ to allow the cells to pre-equilibrate with the XF base medium. An accurate titration with the ATP synthase inhibitor oligomycin was performed for each cell type. Together with the amount of oxygen consumed (OCR) by the cells, values of extracellular acidification (ECAR), which is dependent on mitochondrial-derived $CO_2$

and of glycolysis, were also recorded. Protein quantification was used to normalise the results. OCR and ECAR readings were determined for three cycles (2-min mixing and 5-min measuring), and the baseline measurements were the average of the last three readings before oligomycin addition (1 µM final concentration). The absolute OCR reduction after oligomycin addition was defined at the ATP-dependent OCR (OCR$_{ATP}$). The metabolic phenotype is determined by the OCR$_{ATP}$/ECAR ratios.

### ROS measurement

Cells ($2 \times 10^4$ cells/well) were seeded into 12-well plates and subjected to IACS treatment. Following a 72 h incubation, cells were stained with CellROX (Thermo Fisher Scientific, C10444), and DCFDA (2',7'-Dichlorofluorescin diacetate, D6883, MERCK) and incubated at 37 °C in the dark for 30 min. Fluorescence was measured on a microplate reader at 485 nm excitation and 520 nm emission for CellROX, 485/535 nm Ex/Em for DCFDA. The fluorescent signals were normalised on cell number on a well that underwent the same procedure that was run in parallel.

### Institut Curie clinical cohort

Samples of 503 invasive primary breast cancers excised from patients treated at Institut Curie−Hôpital René Huguenin have been analyzed[63]. All patients cared in our institution before 2007 were informed that their tumour samples might be used for scientific purposes and had the opportunity to decline. After 2007, patients gave their approval by signed inform consent. This study protocol was approved by the local ethical committee (Breast Group of René Huguenin Hospital). Tumour samples were included in the study with a proportion of tumour cells of at least 70%. All patients (mean age 61 years, range 29–91 years) met the following criteria: primary unilateral nonmetastatic breast carcinoma with clinical, histological and biological data were available; no radiotherapy or chemotherapy before surgery; and full follow-up at Institut Curie−Hospital René Huguenin. Modified radical mastectomy was performed in 304 cases (60.4%) and breast-conserving surgery plus locoregional radiotherapy in 196 cases (39%) (data available for 500 patients). The patients had a physical examination and routine chest radiotherapy every 3 months for 2 years, then annually. Adjuvant chemotherapy was administered to 120 patients, hormone therapy was administered to 173 patients and both treatments to 104 patients. The histological classification and the number of positive axillary lymph-nodes were determined at surgery. The Scarff Bloom Richardson's (SBR) histoprognostic system was used to score the malignancy of infiltrating carcinomas.

The status of oestrogen receptor alpha (ERalpha); progesterone receptor (PR) and human epidermal growth factor receptor 2 (ERBB2) was determined at the protein level by using biochemical methods (dextran-coated charcoal method, enzyme immunoassay or immunohistochemistry) and confirmed by real-time quantitative RT-PCR assays[64].

The population was divided into four groups according to hormonal receptors (HR) (ERalpha and PR) and HER2 status, as follows: two luminal subtypes [HR+/HER2+ ($n = 54$)] and [HR+/HER2- ($n = 287$)]; an HER2+ subtype [HR-/HER2+ ($n = 67$)] and a triple-negative subtype [HR−/HER2− ($n = 95$)]. Standard prognostic factors of this tumour set are presented in Table S3. During a median follow-up of 9 years (range 1 month to 33.2 years), 202 patients developed metastasis. Fourteen samples of adjacent normal breast tissue from breast cancer patients and normal breast tissue from women undergoing cosmetic breast surgery were used as sources of normal RNA.

### RNA extraction, cDNA synthesis and PCR amplification

Total RNA was extracted from breast specimens by using the acid-phenol guanidium[64]. RNA was reverse transcribed in a final volume of 20 µl containing 1× RT buffer (500 mM each dNTP, 3 mM MgCl2,

75 mM KCl, 50 mM Tris-HCl pH 8.3), 10 units of RNasinTM ribonuclease inhibitor (Promega, Madison, #N2511, WI, USA), 10 mM dithiothreitol, 50 units of Superscript II RNase H- reverse transcriptase (Gibco-BRL, #18064014, Gaithersburg, MD, USA), 1.5 mM random hexamers (Pharmacia, Uppsala, Sweden) and 1 µg of total RNA. The samples were incubated at 20 °C for 10 min and 42 °C for 30 min, and reverse transcriptase was inactivated by heating at 99 °C for 5 min and cooling at 5 °C for 5 min.

The quality of the RNA samples was determined by electrophoresis through agarose gels and staining with ethidium bromide, and the 18S and 28S RNA bands were visualised under ultraviolet light.

All PCR reactions were performed using QuantStudio 7 Flex real-time PCR system (Perkin-Elmer Applied Biosystems). PCR was performed using the Power SYBR® Green Master Mix (Perkin-Elmer Applied Biosystems, # 4368708). The thermal cycling conditions comprised an initial denaturation step at 95 °C for 10 min and 50 cycles at 95 °C for 15 s and 65 °C for 1 min. Experiments were performed with duplicates for each data point.

Gene expression levels were normalised on the basis of TBP contents (Genbank accession number NM_003194) used as an endogenous RNA control. The expression values of the tumour samples were subsequently normalised such that the median of the expression values of 14 normal breast tissue samples was one.

Primers were designed with the assistance of the Oligo 6.0 computer programme (National Biosciences, Plymouth, MN). The sequences are provided in Table S5.

### Statistical analysis

The distributions of target mRNA levels were characterised by their median values and ranges. Relationships between mRNA levels of the different target genes, and between mRNA levels and clinical parameters, were identified using nonparametric tests, namely the chi-square test (relation between two qualitative parameters), the Mann−Whitney $U$-test (relation between one qualitative parameter and one quantitative parameter) and the Spearman rank correlation test (relation between two quantitative parameters). Differences were considered significant at confidence levels greater than 95% ($p < 0.05$).

To visualise the efficacy of $MRPS12$ and $NDFUS6$ to discriminate two populations (patients that developed/did not develop metastases) in the absence of an arbitrary cut-off value, data were summarized in an ROC (receiver operating characteristic) curve[65]. The AUC (area under curve) was calculated as a single measure for discriminate efficacy. Metastasis-free survival (MFS) was determined as the interval between initial diagnosis and detection of the first metastasis. Survival distributions were estimated by the Kaplan−Meier method, and the significance of differences between survival rates were ascertained with the log-rank test.

### Reporting summary

Further information on research design is available in the Nature Portfolio Reporting Summary linked to this article.

## Data availability

Affymetrix CEL files and normalised log2 RMA data are available at the GEO database under accession no GSE146661. RNAseq raw data have been submitted to the EGA depositary under the following accession codes: EGAS00001006428 EGAS00001006429 EGAS00001006908 Raw data of targeted DNA sequencing have been submitted to the EGA depositary under the number EGAS00001007244. Metabolomic raw data have been deposited in the Metabolights database under the accession code MTBLS3342. Source data related to metabolomic analyses are included in Supplementary Data 1 (comparisons of primary tumours and PDX) and Supplementary Data 2 (comparison of HBCx-124 and HBCx-137 control and IACS-treated xenografts). Normalised metabolomics data and chemical annotations of metabolites

are provided in Supplementary Data 3. Source data related to in vitro and in vivo experiments are provided in the source data file. Source data are provided in this paper.

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

## Acknowledgements

E. Ma. is supported by SIRIC2 (INCa-DGOS-Inserm_12554), GEFLUC and RUBAN ROSE associations for funding. A.M. is supported by AIRC and Fondazione Cassa di Risparmio di Firenze (IG22941 and MultiUser19515) and N.L. is an AIRC fellow. Icons of women, vertebrae and mice were made by Freepik from www.flaticon.com.

## Author contributions

Conceptualisation: F.M.-G., K.D., I.B., A.M. and E.Ma. Methodology: R.E.B., L.M., E.Mo, Z.T., A.S., M.B., N.L., L.H., A.D., P.P., H.D., S.V., J.M.P., V.R., S.B., T.L., A.V.-S., G.D., P.C., G.G., F.M.G., S.H. and K.D. Supervision and Writing—original draft: E.M. Writing—review & editing: E.M., I.B., A.M., S.H., K.D. and Z.K.

## Competing interests

The authors declare no competing interests.

## Additional information

[1]Laboratory of Preclinical Investigation, Translational Research Department, Institut Curie, PSL University, 26 rue d'Ulm, 75005 Paris, France. [2]Department of Genetics, Institut Curie, PSL University, 26 rue d'Ulm, 75005 Paris, France. [3]Dept. of Experimental and Clinical Biomedical Sciences, Viale Morgagni, 50 - 50134, Florence, Italy. [4]ICGex - NGS platform, Institut Curie, PSL University, 26 rue d'Ulm, 75005 Paris, France. [5]INRA, APEX-PAnTher, Oniris, 44322 Rue de la Géraudière, Nantes, France. [6]Department of Pathology, Institut Curie, PSL University, 26 rue d'Ulm, 75005 Paris, France. [7]Department of Surgery, Institut Curie, PSL University, 26 rue d'Ulm, 75005 Paris, France. [8]Department of Medical Oncology, Institut Curie, PSL University, 26 rue d'Ulm, 75005 Paris, France. [9]"Stress and Cancer" Laboratory, Institut Curie - Inserm U830, PSL University, 26 rue d'Ulm, 75005 Paris, France. [10]Metabolon Inc., 617 Davis Drive, Suite 100, Morrisville, NC 27560, USA. [11]Paris City University, Inserm U1016, Faculty of Pharmaceutical and Biological Sciences, 75005 Paris, France. ✉e-mail: elisabetta.marangoni@curie.fr

