## [Peer Review File · Nature Communications]

REVIEWER COMMENTS

Reviewer #1 (Remarks to the Author):

In their manuscript Botty et al report the impact of inhibiting Complex I (IACS small molecule inhibitor) in breast cancer PDX models. They perform transcriptomic and metabolomic analyses of IACS responder and non-responder PDX models and metabolomics upon IACS treatment in vitro. The authors find a correlation between IACS sensitivity and resistance to hormone and CDK4/6i therapy and metastasis. IACS sensitivity is related to the expression of OXPHOS genes and the ability to activate glycolysis upon IACS treatment. The authors propose that the information gleaned from these analyses will guide future treatment with IACS or other Complex I inhibitors in breast cancer patients. Overall, the analyses are performed with a high degree of experimental rigor and the effects on tumor growth and differential effects in sensitive and resistant models is impressive. I agree with the authors that these data will provide important insight into inhibition of Complex I in future cancer treatments.

Major Comments

1. In the abstract, the author state, "At the metabolic level, in vivo response to IACS-010759 was associated with inhibition of glutathione, glycogen and pentose metabolism." The data collected (steady state metabolite levels) are not sufficient to infer inhibition of the corresponding metabolic pathways. The authors would need to perform experiments using labeled metabolites to make these claims. Adjusting the language to more accurately describe the data presented would avoid the costly/complex experiments needed to measure pathway activity.

Minor comments

1. In Figure 1E, it is unclear what conclusions the authors are trying to draw, as the parent palbo sensitive tumor is already IACS sensitive. Also, it is unclear if there is an additive or synergistic effect of combination treatment from the data provided.
2. Authors state, "amplification of MYC, which is known to activate OXPHOS by regulating the transcription of mitochondrial genes"
This statement is a bit inaccurate as MYC amplification activates numerous metabolic pathways, including all of glycolysis.
3. Authors state, "The levels of several gamma-glutamyl amino acids, which are intermediates in glutathione metabolism"
Gamma-glutamyl amino acids are, strictly speaking, not intermediates in glutathione metabolism. They can be produced as a byproduct of glutathione degradation or in the gamma-glutamyl cycle which is predominantly a salvage/uptake pathway.
4. Authors state, "the metabolites levels of TCA cycle, which is tightly coordinated with OXPHOS" in which they seem to imply that the levels of the TCA cycle metabolites are indicative of OXPHOS activity. However, there is not much evidence for correlation between TCA cycle intermediate levels and OXPHOS function in different cancer states.
5. Do the authors know at what concentration of IACS off-target effects begin to impact cell viability? One might imagine that pyruvate+uridine would rescue on-target effects but not off-target effects. As such, the high doses of IACS used in Figure 3 might not be relevant to their analysis (and may make these data indicate an even stronger relationship).
6. Can the authors clarify in the legend/text whether the RNAseq data from Figure 4 was performed on tumors not treated with IACS.
7. The results of Figure 5E is not possible to interpret without a BSO alone condition.
8. Reductions in aspartate levels are commonly observed under OXPHOS inhibition (see PMID 26232224, 29941933, 26232225).

Reviewer #2 (Remarks to the Author):

The authors have conducted a series of in vitro and in vivo experiments in ER+ metastatic breast cancer models to test their dependence on OXPHOS.

By analyzing a set of published gene expression data from PDXs samples derived from BC bone metastases, they identify OXPHOS as one of the pathways enriched in the BMs compared to

matched primary tumors.

By metabolomic analysis of 6 PDXs and 6 PTs (4 of which matched), they also saw significant difference in metabolites involved in diverse pathways, including the TCA cycle, GSH metabolism intermediates, hexosamine and purine/pyrimidine metabolism.

They then tested the in vivo activity of a potent and selective OXPHOS inhibitor, IACS-10759, in 7 ER+ bone mets PDX (5 from the previous metabolomic analysis. 5 of these responded to treatment to a variable degree, while 2 others were resistant. They also evaluated response to an ER- liver met PD and two from ER+ primary tumors and all were resistant to treatment

By testing a published palbociclib-resistant model compared to its sensitive counterpart, they also saw remarkable sensitivity of the palbociclib-resistant model to IACS.

Testing of 3 isogenic lines, normal control MCF7, MCF7 made resistant to estrogen deprivation or to palbociclib, with IACS showed that resistant cells were much more sensitive to the inhibitor compared to the parental line. Sensitive lines had a higher OCR/ECAR ratio, and this could explain their enhanced sensitivity to IACS.

The author also tested the activity of IACS in two PDX derived from bone mets of patients who had shown progressive disease treated with palbociclib and letrozole. Both responded to a variable degree to the administration of IACS alone, but treatment with fulvestrant and IACS resulted in tumor regression.

By combined analysis of all the PDX tested, the authors concluded that MYC amplification (1 PDX) and PI3K/AKT mutations (4 PDX) were associated with response to IACS treatment. By GSEA analysis comparing gene expression in IACS sensitive and resistant models, the authors identified interferon, estrogen, PI3K and OXPHOS signatures as markers of response, while proliferation and EMT pathway signatures seemed to be associated with resistance.

The authors then further characterized one IACS- resistant and one sensitive PDX, by performing an unbiased metabolomic analysis before and after treatment. This highlighted significant change in both after treatment (FA and branched chain amino acids metabolism) and acyl carnitines utilization. There seemed to be an increase of FA content at baseline in the resistant PDX compared to the sensitive one. GSH and SAM levels (and related metabolites), as well as ascorbate metabolism metabolites were higher in the resistant one. The authors indeed found that combined treatment with IACS and BSO, enhanced the activity of the compound in the resistant PDX tested ex vivo. ROS levels were also induced by treatment as well as changes in TCA cycle intermediates in both the resistant and the sensitive model. Surprisingly, the authors found that levels of glycolytic intermediates were not induced more in the resistant model, while they observed a significant difference in pentose pathway and pentose metabolism intermediates, as well as in glycogen metabolism.

Analysis of a cohort of 503 breast cancer tumor samples highlighted a negative prognostic correlation with survival of one OXPHOS pathway gene, MRPS12 in RH+ and triple negative cases, and another, NDFUS6 in RH+ ones. Expression of both genes, correlated with proliferative index and histological grade and ER and PR status.

This study provides a comprehensive tour de force in the analyses of metabolic connotations of patient-derived breast cancer bone metastasis samples, correlating response to treatment with an OXPHOS inhibitor with gene expression and metabolic profiling, in single agents and combination treatment.

While recognizing the quality of the study, I am left with the question of relevance and translatability, with the study highlighting the complexity of what might be driving response to OXPHOS inhibitors. It seems also that many studies have now been published on this subject. Furthermore, the path towards clinical development of such inhibitors seems unclear.

Reviewer #3 (Remarks to the Author):

The authors first identify increased oxidative phosphorylation as a feature characterizing ER+ metastasis vs primary tumors using multiple pdx models. IACS an inhibitor of complex I could reduce the growth of these pdx models including one resistant to palbociclib. The development, characterization and treatment of these models is of critical importance to understanding ER+

metastatic cancers resistant to endocrine therapy and cdk4/6i. They have provocative data which identifies ways to target the cancer.

The data are sound and did not identify any significant concerns.

I have minor concerns listed below.

1. However, they focus on only 2 lines from patients who were treated with cdk4/6i and would recommend that they treat the PDX lines from mets and develop at least one or two other lines of palbo/fulvestrant resistance.

2. Additionally, the lines they generated are not quite appropriate. Specifically: you didn't pick samples that were clearly cdk4/6i resistant?

Case 227 vertebral metastasis was done prior to palbo/letrozole treatment thus not resistant

Case 180 vertebral metastasis was done after taxol but not immediately upon resistance to palbo/letrozole.

2. Did you consider other complex I inhibitors that can be used clinically including metformin, atovaquone, or arsenic trioxide? Notably, metformin has NOT been effective in the treatment of patients with metastatic disease.

3. "To identify metabolic changes associated to response or resistance to IACS-010759, we performed a global metabolomic analysis of untreated and treated xenografts from a IACS responder (HBCx-124) and a IACS resistant PDX (HBCx-137)." Why did you only do a comparison on two cell lines. You have more resistant vs sensitive lines yet you only did these two comparisons?

They also showed that endocrine and palbociclib resistant cells had increased ROS levels upon inhibition of OXPHOS with IACS.

Finally, in ER+ BC patients, high expression of several OXPHOS associated genes predicted poor prognosis.

1. Could you also analyze expression of these in biopsies of patient samples with resistance to endocrine therapy and endocrine/CDK4/6i therapy?

RESPONSE TO REVIEWERS' COMMENTS

REVIEWER COMMENTS

Reviewer #1 (Remarks to the Author):

In their manuscript Botty et al report the impact of inhibiting Complex I (IACS small molecule inhibitor) in breast cancer PDX models. They perform transcriptomic and metabolomic analyses of IACS responder and non-responder PDX models and metabolomics upon IACS treatment in vitro. The authors find a correlation between IACS sensitivity and resistance to hormone and CDK4/6i therapy and metastasis. IACS sensitivity is related to the expression of OXPHOS genes and the ability to activate glycolysis upon IACS treatment. The authors propose that the information gleaned from these analyses will guide future treatment with IACS or other Complex I inhibitors in breast cancer patients. Overall, the analyses are performed with a high degree of experimental rigor and the effects on tumor growth and differential effects in sensitive and resistant models is impressive. I agree with the authors that these data will provide important insight into inhibition of Complex I in future cancer treatments.

Major Comments

1. In the abstract, the author state, "At the metabolic level, *in vivo* response to IACS-010759 was associated with inhibition of glutathione, glycogen and pentose metabolism." The data collected (steady state metabolite levels) are not sufficient to infer inhibition of the corresponding metabolic pathways. The authors would need to perform experiments using labeled metabolites to make these claims. Adjusting the language to more accurately describe the data presented would avoid the costly/complex experiments needed to measure pathway activity.

The sentence in the abstract was adjusted as follow:

At the metabolic level, *in vivo* response to IACS-010759 was associated with decreased levels of glutathione, glycogen and pentose metabolites in treated tumors.

The language was adjusted through all the manuscript.

Minor comments

1. In Figure 1E, it is unclear what conclusions the authors are trying to draw, as the parent palbo sensitive tumor is already IACS sensitive. Also, it is unclear if there is an additive or synergistic effect of combination treatment from the data provided.

The conclusions of this experiment are 1) that IACS treatment is also efficient in tumours with acquired resistance to CDK4/6 inhibitors 2) that acquired resistance to CDK4/6 does not impact the response to IACS

In the HBCx-124 palboR model (figure 2 F) The difference between IACS and IACS + palbo fulv was not statistically significant, although the median survival was greater (126 as compared to 144 days).

By contrast, in the HBCx-180 and HBCx-227 models (figure 3C) the combination was more efficient than IACS and palbo-fulv monotherapy groups. However, these *in vivo* data are not sufficient to determine whether the effect was synergistic or additive (to assess this, different combinations groups with different doses should be tested).

2. Authors state, “amplification of MYC, which is known to activate OXPHOS by regulating the transcription of mitochondrial genes”

This statement is a bit inaccurate as MYC amplification activates numerous metabolic pathways, including all of glycolysis.

The sentence was modified:

Interestingly, the PDX with the highest response to IACS (HBCx-124) (Figure 2B) harbors an amplification of MYC, a regulator of biosynthetic and metabolic pathways including OXPHOS^{21, 22}.

3. Authors state, “The levels of several gamma-glutamyl amino acids, which are intermediates in glutathione metabolism”

Gamma-glutamyl amino acids are, strictly speaking, not intermediates in glutathione metabolism. They can be produced as a byproduct of glutathione degradation or in the gamma-glutamyl cycle which is predominantly a salvage/uptake pathway.

The sentence was modified as follow (lines 103-104)

The levels of several gamma-glutamyl amino acids, **that play an important role in glutathione homeostasis through the gamma-glutamyl cycle**, are shown in Figure 1E and are significantly enriched in BM PDX.

4. Authors state, “the metabolites levels of TCA cycle, which is tightly coordinated with OXPHOS” in which they seem to imply that the levels of the TCA cycle metabolites are indicative of OXPHOS activity. However, there is not much evidence for correlation between TCA cycle intermediate levels and OXPHOS function in different cancer states.

The sentence was modified as follow (lines 105-109)

Moreover, the levels of the relevant TCA intermediates are also enriched as shown in Figure 1F in BM PDX. Although the levels of the increased metabolites revealed by the steady state analysis do not imply *per se* enhancement of the TCA flux, an overall increase in the TCA cycle intermediates could contribute to the production of reducing equivalents (NADH+ H+ and FADH₂) and subsequent activity of the mitochondrial electrons transport respiratory chain¹⁴.

5. Do the authors know at what concentration of IACS off-target effects begin to impact cell viability? One might imagine that pyruvate+uridine would rescue on-target effects but not off-target effects. As such, the high doses of IACS used in Figure 3 might not be relevant to their analysis (and may make these data indicate an even stronger relationship).

This is an interesting question. IACS is supposed to be one of the more potent and specific inhibitors of complex I. Different publications report *in vitro* experiments performed to rescue on target effects of oxphos inhibitors (including IACS), with pyruvate and uridine but also with succinate. Performing these experiments *in vivo* is much more difficult. Succinate prodrugs has a too short half-life in rodents (personal communication of Pr. Johannes Ehinger, who has a long expertise on mitochondrial dysfunction and rescue experiments).

High doses of uridine (3500 mg/kg) have been shown to partially rescue 5-FU related toxicities in different mice strain, however the effects were associated with severe hypothermia (Codanacci-Pisanelli, 1997; Peters et al. 1987).

Despite this, we performed a first experiment with lower dose of uridine and pyruvate (2000 mg/kg/day and 500 mg/kg/day, respectively), that we associated with the 10 mg/kg dose of IACS in the PDX with the highest response to IACS. These doses did not change the response to IACS.

Figure 1

In this experiment, mice body temperature was monitored every day and was found to be unchanged.

Therefore, we decided to repeat the experiment with higher doses of uridine and pyruvate (3000 mg/kg/day and 800 mg/kg/day). In this experiment, supplementation of uridine + pyruvate decreased tumour growth, but the combination with IACS did not change the response.

By calculating the % of tumor growth inhibition of IACS + U + P / U + P as compared to IACS/control it appears that the TGI is decreased during the two first weeks of treatments. However, this is due to the anti-tumor activity of U+P and the response in the combination group does not appear to be rescued.

In this experiment, we observed a mild decrease of body temperature after 2 weeks of treatment, 1 h after uridine treatment but that was resolved 5 h after.

N° Souris	Tps	Body T° (°C)														
		D1	D2	D3	D4	D5	D6	D7	D8	D9	D10	D11	D12	D13	D14	D15
69	Before TT	/	37,0°C	/	/	36,5°C	36,8°C	36,5°C	36,8°C	36,9°C	/	/	36,5°C	36,8°C	36,8°C	36,8°C
	1h post TT	/	36,6°C	/	/	36,4°C	36,6°C	36,5°C	36,6°C	37,4°C	/	/	36,5°C	36,5°C	33,7°C	34,7°C
	5h post TT	/	36,4°C	/	/	36,8°C	36,4°C	36,4°C	36,7°C	37,0°C	/	/	36,5°C	36,4°C	36,6°C	36,6°C
53	Before TT	/	36,8°C	/	/	36,3°C	37,1°C	36,6°C	36,3°C	37,4°C	/	/	36,8°C	36,7°C	37,2°C	36,8°C
	1h post TT	/	36,5°C	/	/	36,4°C	36,6°C	36,4°C	36,2°C	37,1°C	/	/	36,2°C	36,8°C	32,8°C	34,3°C
	5h post TT	/	36,4°C	/	/	36,5°C	37,2°C	36,4°C	36,5°C	37,1°C	/	/	36,5°C	36,6°C	36,8°C	37,1°C
26	Before TT	/	36,7°C	/	/	37,2°C	37,3°C	36,5°C	36,5°C	36,8°C	/	/	36,7°C	36,6°C	37,1°C	36,7°C
	1h post TT	/	37,0°C	/	/	36,9°C	36,6°C	36,7°C	36,5°C	37,0°C	/	/	36,6°C	37,0°C	36,4°C	35,4°C
	5h post TT	/	36,6°C	/	/	36,8°C	36,5°C	36,9°C	36,5°C	37,0°C	/	/	36,7°C	36,5°C	36,5°C	36,6°C
67	Before TT	/	36,9°C	/	/	36,4°C	36,7°C	36,5°C	36,5°C	37,1°C	/	/	36,9°C	37,6°C	36,7°C	36,8°C
	1h post TT	/	36,3°C	/	/	36,2°C	36,5°C	33,7°C	33,1°C	37,2°C	/	/	36,2°C	32,8°C	31,9°C	32,7°C
	5h post TT	/	36,7°C	/	/	36,3°C	36,6°C	36,3°C	36,3°C	37,2°C	/	/	37,2°C	36,2°C	36,7°C	36,5°C
55	Before TT	/	36,5°C	/	/	36,4°C	36,5°C	36,3°C	36,3°C	37,2°C	/	/	36,4°C	36,5°C	36,8°C	36,8°C
	1h post TT	/	37,0°C	/	/	36,3°C	36,8°C	36,7°C	36,1°C	37,2°C	/	/	36,3°C	36,5°C	32,7°C	33,1°C
	5h post TT	/	36,3°C	/	/	36,2°C	36,5°C	36,8°C	36,3°C	37,2°C	/	/	36,6°C	36,6°C	36,5°C	36,5°C
49	Before TT	/	36,8°C	/	/	36,5°C	36,8°C	36,8°C	36,3°C	36,9°C	/	/	36,6°C	36,7°C	37,2°C	36,7°C
	1h post TT	/	36,6°C	/	/	36,4°C	37,2°C	36,3°C	36,2°C	36,8°C	/	/	36,7°C	37,1°C	35,1°C	35,4°C
	5h post TT	/	36,5°C	/	/	36,3°C	36,5°C	36,6°C	36,4°C	36,8°C	/	/	37,1°C	36,5°C	36,5°C	36,5°C

Based on these results, we decided not to include these experiments in the manuscript. We added a sentence in the discussion about the possibility of off-target effects in the responder tumors.

Page 18 (line 407)

Second, we cannot exclude the existence of off-target effects of the OXPLOS inhibitor dosed at 10 mg/kg in responder models.

Furthermore, to analyze more in depth the biological pathways altered by IACS treatment, we performed an RNAseq analysis of HBCx-124 xenografts (the most responder PDX) after 5 days of treatment. These results have been added in Figure 2C and Table S2.

The GSEA analysis of RNAseq data identified downregulation of different gene sets associated with cell proliferation, E2F targets, G2M checkpoint, mitosis in IACS-treated xenografts, while up-regulated gene sets were associated with activation of the mRNA upon binding of the cap binding complex, translation initiation and cell death. These results are concordant with those reported by Molina et al. (PMID: 29892070) in different cell lines treated by IACS.

Concerning comment about the high dose of 10 mg/kg, we tested two lower doses, 5 mg/kg and 2.5 mg/kg in 2 responder models. In both models, the 5 mg/kg dose was still efficient, resulting in stable disease, while the dose of 2.5 mg/kg showed a modest tumor growth inhibition as compared to control. These results have been added in figure 2B and supplementary figure S1.

6. Can the authors clarify in the legend/text whether the RNAseq data from Figure 4 was performed on tumors not treated with IACS.

RNAseq data of figure 4 were generated from untreated xenografts (added in the legend and text).

7. The results of Figure 5E is not possible to interpret without a BSO alone condition.

Figure 5E was modified as followed

8. Reductions in aspartate levels are commonly observed under OXPHOS inhibition (see PMID 26232224, 29941933, 26232225).

This was specified in a sentence at page 12, line 240:

The levels of aspartate, whose synthesis is supported by electron acceptors provided by mitochondrial respiration²⁵ were strongly reduced in both responder and resistant PDX after IACS treatment (Figure 6B).

Reviewer #2 (Remarks to the Author):

The authors have conducted a series of in vitro and in vivo experiments in ER+ metastatic breast cancer models to test their dependence on OXPHOS.

By analyzing a set of published gene expression data from PDXs samples derived from BC bone metastases, they identify OXPHOS as one of the pathways enriched in the BMs compared to matched primary tumors.

By metabolomic analysis of 6 PDXs and 6 PTs (4 of which matched), they also saw significant difference in metabolites involved in diverse pathways, including the TCA cycle, GSH metabolism intermediates, hexosamine and purine/pyrimidine metabolism.

They then tested the in vivo activity of a potent and selective OXPHOS inhibitor, IACS-10759, in 7 ER+ bone mets PDX (5 from the previous metabolomic analysis). 5 of these responded to treatment to a variable degree, while 2 others were resistant. They also evaluated response to an ER- liver met PD and two from ER+ primary tumors and all were resistant to treatment.

By testing a published palbociclib-resistant model compared to its sensitive counterpart, they also saw remarkable sensitivity of the palbociclib-resistant model to IACS.

Testing of 3 isogenic lines, normal control MCF7, MCF7 made resistant to estrogen deprivation or to palbociclib, with IACS showed that resistant cells were much more sensitive to the inhibitor compared to the parental line. Sensitive lines had a higher OCR/ECAR ratio, and this could explain their enhanced sensitivity to IACS.

The author also tested the activity of IACS in two PDX derived from bone mets of patients who had shown progressive disease treated with palbociclib and letrozole. Both responded to a variable degree to the administration of IACS alone, but treatment with fulvestrant and IACS resulted in tumor regression.

By combined analysis of all the PDX tested, the authors concluded that MYC amplification (1 PDX) and PI3K/AKT mutations (4 PDX) were associated with response to IACS treatment. By GSEA analysis comparing gene expression in IACS sensitive and resistant models, the authors identified interferon, estrogen, PI3K and OXPHOS signatures as markers of response, while proliferation and EMT pathway signatures seemed to be associated with resistance.

The authors then further characterized one IACS-resistant and one sensitive PDX, by performing an unbiased metabolomic analysis before and after treatment. This highlighted significant change in both after treatment (FA and branched chain amino acids metabolism) and acyl carnitines utilization. There seemed to be an increase of FA content at baseline in the resistant PDX compared to the sensitive one. GSH and SAM levels (and related metabolites), as well as ascorbate metabolism metabolites were higher in the resistant one. The authors indeed found that combined treatment with IACS and BSO, enhanced the activity of the compound in the resistant PDX tested ex vivo. ROS levels were also induced by treatment as well as changes in TCA cycle intermediates in both the resistant and the sensitive model. Surprisingly, the authors found that levels of glycolytic intermediates were not induced more in the resistant model, while they observed a significant difference in pentose pathway and pentose metabolism intermediates, as well as in glycogen metabolism.

Analysis of a cohort of 503 breast cancer tumor samples highlighted a negative prognostic correlation with survival of one OXPHOS pathway gene, MRPS12 in RH+ and triple negative cases, and another, NDFUS6 in RH+ ones. Expression of both genes, correlated with proliferative index and histological grade and ER and PR status.

This study provides a comprehensive tour de force in the analyses of metabolic connotations of patient-derived breast cancer bone metastasis samples, correlating response to treatment with an OXPHOS inhibitor with gene expression and metabolic profiling, in single agents and combination treatment.

While recognizing the quality of the study, I am left with the question of relevance and translatability, with the study highlighting the complexity of what might be driving response to OXPHOS inhibitors. It seems also that many studies have now been published on this subject. Furthermore, the path towards clinical development of such inhibitors seems unclear.

It is true that many studies have been published on complex I targeting, however, to date and to our knowledge in other cancer types or in TNBC. This study is the first to demonstrate that OXPHOS is a metabolic vulnerability in treatment resistant advanced ER+ breast cancer. However, we agree that the path towards clinical development is not as straightforward as appeared in the preclinical studies. We have addressed the question of translatability and clinical development in the discussion (page 19, lines 416-426).

Reviewer #3 (Remarks to the Author):

The authors first identify increased oxidative phosphorylation as a feature characterizing ER+ metastasis vs primary tumors using multiple pdx models. IACS an inhibitor of complex I could reduce the growth of these pdx models including one resistant to palbociclib. The development, characterization and treatment of these models is of critical importance to understanding ER+ metastatic cancers resistant to endocrine therapy and cdk4/6i. They have provocative data which identifies ways to target the cancer.

The data are sound and did not identify any significant concerns.

I have minor concerns listed below.

1. However, they focus on only 2 line from patients who were treated with cdk4/6 i and would recommend that they treat the pdx lines from mets and develop at least one or two other lines of palbo/fulvestrant resistance.

We agree that having additional PDX models from patients treated with CDK4/6 inhibitors would be highly relevant. Our program of PDX establishment from CDK4/6 treated patients is still ongoing, but unfortunately, we did not establish new models yet.

However, we have tested the efficacy of IACS in an additional model of Palbociclib acquired resistance previously reported (Montaudon et al. Nat Com 2020).

The results, added in figure 2 g, h, i, showed a striking anti-tumor activity of IACS, alone or combined to palbo-fulvestrant, in this model. Results are described at page 7 (lines 162-164).

2. Additionally, the lines they generated are not quite appropriate. Specifically: you didn't pick samples that were clearly cdk4/6/ai resistant?

Yes, the 2 models of figure 3 (HBCx-180 and HBCx-227) were established from patients classified as resistant to palbo + AI (both patients showed progressive disease and had to switch treatment).

The 2 PDX models were also resistant to palbo-fulv (progressive disease after 3-5 weeks of treatments).

Case 227 Vertebroplasty was done prior to palbo/letrozole treatment thus not resistant

It may be an issue of what it is called resistant in this particular case. Indeed, although vertebroplasty was done prior to palbociclib and letrozole treatment in the patients, the PDX originated was resistant, although not previously treated by palbo. One can speculate that this indeed a de novo palbociclib resistant tumors which has been described in 20-30% of the cases in the clinic.

Case 180 vertebroplasty was done after taxol pod not immediately upon resistance to palbo/letrozole.

It's true. The time of vertebroplasty was decided by the clinicians based on medical criteria, not in function of the pdx establishment. However, the PDX retains palbo/letrozole resistance and therefore it is a relevant model in the context of our study.

2. did you consider other complex 1 inhibitors that can be used clinically including metformin, atovaquone, or arsenic trioxide? Notably, metformin has NOT been effective in the treatment of patients with metastatic disease.

To answer to this question, we tested metformin (at high dose) as compared to IACS in a IACS responder PDX, without finding any anti-tumour effect.

The range of metformin doses used for *in vivo* studies is large: but different works suggest that metformin has antineoplastic activity attributable to inhibition of complex I in the micromolar range *in vitro* and at high doses *in vivo* (between 200 and 500 mg/kg/day).

Metformin however influences other mechanisms independently of complex I inhibition (such as fatty acids oxidation) and it is not clear whether the *in vivo* anti tumour effect of this drug (which are modest), reported in other publications, depend on CI inhibition or on these CI independent mechanisms or both.

This result has been added in Figure 2B and is discussed at page 16 lines 350 -355

3. "To identify metabolic changes associated to response or resistance to IACS-010759, we performed a global metabolomic analysis of untreated and treated xenografts from a IACS responder (HBCx-124) and a IACS resistant PDX (HBCx-137)." Why did you only do a comparison on two cell lines. You have more resistant vs sensitive lines yet you only did these two comparisons?

We agree with this comment.

The metabolic analysis of multiple PDX including at least 4-5 replicates for both control and treated tumours was too expensive. An alternative could have been to analyze more PDX and decrease the number of replicates, however we decided to keep at least 5 replicated given the heterogeneity between xenografts from the same PDX.

The fact that the PD analysis was limited to 2 PDX was underlined as a limitation of our study (page 18, line 404).

They also showed that endocrine and palbociclib resistant cells had increased ROS levels upon inhibition of oxphos with IACS.

Finally, in ER+ BC patients, high expression of several OXPHOS associated genes predicted poor prognosis.

1. could you also analyze expression of these in biopsies of patient samples with resistance to endocrine therapy and endocrine/cdk4/6i therapy?

We agree that having access to biopsies from patients responder and resistant to CDK4/6 would add important information to our study. In our hospital, however, we have no access to such a cohort of patients.

Interestingly, a comparative biomarker analysis of paloma 2/3 trials identified Oxidative phosphorylation among the pathways associated with resistance to Palbociclib in both PALOMA 2 and 3 trials (PMID: 35974168).

We contacted several times the principal investigator of these studies asking access to the data to perform a more detailed analysis of oxidative phosphorylation in these trials, however we received no answer.

We cited this paper in the discussion at page 18 (lines 418).

REVIEWERS' COMMENTS

Reviewer #1 (Remarks to the Author):

The authors have addressed all of my concerns.

Reviewer #2 (Remarks to the Author):

I am satisfied with the authors' revision of the manuscript. I am also hopeful about the advancement through the clinic of additional inhibitors, e.g. the novel biguanide IM156. Were this compound to advance further, based on this study, there could be a great opportunity to evaluate response in metastatic breast cancer patients.

Reviewer #3 (Remarks to the Author):

most of my concerns were adequately addressed.

Please comment on one further question/concern which is did you identify INK4A, FAT1, PTEN or ARID1A loss in your screen for palbo resistant cancers (either resistant or sensitive to IACS-010759)? The loss of these tumor suppressors are described features of cdk4/6i resistant disease.

RESPONSE TO REVIEWERS' COMMENTS

Reviewer #1 (Remarks to the Author):

The authors have addressed all of my concerns.

Reviewer #2 (Remarks to the Author):

I am satisfied with the authors' revision of the manuscript. I am also hopeful about the advancement through the clinic of additional inhibitors, e.g. the novel biguanide IM156. Were this compound to advance further, based on this study, there could be a great opportunity to evaluate response in metastatic breast cancer patients.

Reviewer #3 (Remarks to the Author):

most of my concerns were adequately addressed.

Please comment on one further question/concern which is did you identify INK4A, FAT1, PTEN or ARID1A loss in your screen for palbo resistant cancers (either resistant or sensitive to IACS-010759)? The loss of these tumor suppressors are described features of cdk4/6i resistant disease.

Response:

Thank you for this suggestion, we did not find loss of these genes in our palbo-resistant models.